# Comparative analysis and FPGA realization of different control synchronization approaches for chaos-based secured communication systems

Talal Bonny[1]*, Wafaa Al Nassan[1], Aceng Sambas[2,3]

**1** Department of Computer Engineering, University of Sharjah, Sharjah, United Arab Emirates, **2** Faculty of Informatics and Computing, Universiti Sultan Zainal Abidin, Gong Badak, Terengganu, Malaysia, **3** Department of Mechanical Engineering, Universitas Muhammadiyah Tasikmalaya, Tasikmalaya, Jawa Barat, Indonesia

* tbonny@sharjah.ac.ae

**Data Availability Statement:** All relevant data are within the paper and its Supporting Information files.

## Abstract

Synchronization of the chaotic systems has attracted much attention in recent years due to its vital applications in secured communication systems. In this paper, an implementation and comparative analysis of two different control approaches for synchronization between two identical four-dimensional hyperchaotic systems is presented. The two control approaches are the Adaptive nonlinear controller and the linear optimal quadratic regulator LQR. To demonstrate the effectiveness of each controller, the numerical simulation is presented using Matlab/Simulink and the control law is derived. The performance of the proposed controllers is compared based on four factors; response time, squared error integration, energy applied from the controller, and cost function. To measure the robustness of the control approaches, the performance factors are compared when there is a change in system parameters and a variation in the initial conditions. Then the proposed synchronization methods are implemented on the FPGA platform to demonstrate the utilized resources on Field Programmable Gate Array (FPGA) hardware platform and the operation speed. Finally, to generalize the results of the comparison, the study is implemented for the synchronization of another secured communication system consisting of two identical three-dimensional chaotic. The experimental results show that the LQR method is more effective than the Adaptive controller based on the performance factors we propose. Moreover, the LQR is much simpler to implement on hardware and requires fewer resources on the FPGA.

## 1. Introduction

Chaotic oscillators have been involved in many research fields and applications such as robotics [1, 2], secured communication [3, 4], cryptography [5], neural network [6], etc. In 1963, Lorenz found a 3-D (three-dimensional) weather modeling system, which resulted in the

**Funding:** The author(s) received no specific funding for this work.

**Competing interests:** The authors have declared that no competing interests exist.

establishment of the first chaotic system [7]. Since then, chaos theory has been studied intensively by scientists and engineers, which has been applied in various fields [8–10]. In 1979, a new type of chaotic system appeared when Rossler introduced a 4-D system named a hyperchaotic system which is a chaotic system with more than one positive Lyapunov exponent [11]. Hyperchaotic systems have been the subject of extensive research due to there intriguing characteristics, including high capacity, high security, and high efficiency [12].

Many chaotic/hyperchaotic systems have been introduced in both continuous-time and discrete-time. Chua's circuit, Duffing, Rossler, Lu and Chen as continuous-time systems. Logistic map, Henon map, Henon 3-D map and Lorenz discrete map are discrete-time systems. In addition, the chaotic systems presented in previous work have been implemented using different techniques such as electronic circuits [13, 14], FPGA implementation [15–18] and memristor-based circuits [19].

In 1993, Cuomo et al. [20, 21] introduced the first secure communication system using chaotic systems. In this system the chaotic signal was used to transmit data from the transmitter to the receiver since chaotic signals can be used to conceal messages because they are aperiodic, broadband, and have a wide spectrum. Ye et al. [22] proposed a circuit model with multiple memristors based on the complete RLC structure and implemented for image encryption. They show that the encryption algorithm based on this mixed memristive chaotic system has higher security and better anti-decoding ability. Moreover, a hyperchaotic memristive circuit based on Wien-bridge chaotic circuit was designed [23]. Liu et al. [24] presented of the new 3D chaotic system with many coexisting attractors and its application to weak signal detection. Also, the proposed system can output more key information as a pseudo-random signal generator (PRSG). Xu et al. [25] proposed a new fractional-order chaotic system based on the model of 4- neurons-based Hopfield Neural Network (HNN) using Adomain decomposition method and implemented a new image encryption algorithm based on the multiple hash index chain. Furthermore, Sambas et al. [26] presented of the new 3D chaotic system with line equilibrium and it's application in image encryption. They also display the implementation of the Field-Programmable Gate Array (FPGA) based Pseudo-Random Number Generator (PRNG) by using the new chaotic system.

One of the important challenges in designing a chaos-based secured communication system is the synchronization between the transmitter and the receiver oscillations when the initial conditions change. Because chaotic systems show high sensitivity to initial conditions, so the synchronization of chaotic systems seems to be impossible till the working of Pecora and Carrol in 1990. They showed that if two chaotic systems linked together by a common signal, it is possible to obtain chaotic synchronization regardless of the initial conditions [27].

Synchronization of many pairs of chaotic systems using various control techniques have been studied in the previous work such as backstepping control [28], nonlinear active control [29], sliding mode synchronization [30, 31], adaptive control [32, 33], impulsive control [34], optimal control [35], PID [36], fuzzy logic control [37] and ANN [38]. In real-time, chaotic synchronization is required for successful message recovery [39] because of uncertainty, disturbance and the sensitivity to initial conditions. The mostly used synchronization technique is adaptive synchronization. Several investigations studied chaotic synchronization using Adaptive control.

The article [40] proposed a novel adaptive nonlinear control scheme for asymptotic chaos synchronization of controlled hyperchaotic Chen chaotic systems with five uncertain parameters. In [41], the adaptive chaos synchronizations between the hyperchaotic LS system and CYQY system, and between the LS system and hyperchaotic Chen system have been studied. In [42], the backstepping control method has been applied to achieve global chaos hybrid synchronization for the n-scroll Chua and Lur'e chaotic systems. The nonlinear active control

method was used in [43] to achieve hybrid chaos synchronization of identical Arneodo chaotic systems, identical Rossler chaotic systems and non-identical Arneodo and Rossler chaotic systems. Two kinds of sliding mode synchronization problems of multiple chaotic systems have been addressed in [44], by considering two different synchronization modes. New synchronization criteria were given by selecting suitable sliding mode manifolds and designing sliding mode controllers and adaptive laws.

Most of the control methods yield good results and achieve the synchronization between the master and the slave systems. However, there is no previous contribution that differentiates between the control methods and selects the one which achieves the best synchronization performance when the oscillator parameters or the initial conditions change.

To the best of our knowledge, we are the first who study different control synchronization methods for chaos-based communication systems and show the control method that gives the best performance when there is uncertainty in the system parameters or a variation in the initial conditions. Moreover, we used hardware technique developed in [17, 18, 45, 46] to design both synchronization systems using Field Programmable Gate Array (FPGA) platform.

In this paper we decided to use digital implementation using FPGA, even though we appreciate the value in using analog implementation using Field-Programmable Analog Array FPAA [47]. Digital tools allow us to repeat experiments more accurately, which is important when we want to see how our control method interacts with nonlinear systems. Digital systems let us compare different ways of controlling things directly, so we can see how well our approach works compared to other methods. The clarity of documentation and data recording in digital implementations aids transparent and replicable reporting of our methodology and results.

In this work, our secured communication system is built using two identical 4-D hyperchaotic oscillators with hidden attractors. The two control methods used for synchronization are the Adaptive control and non-adaptive Linear Quadratic Regulator (LQR) methods. The performance indicators for the evaluation of the studied control methods are applied for two different types of chaotic oscillators; high dimensional chaotic system and low dimensional one.

The main contributions in this work are summarized as follows:

1. Proposing a chaos-based secured communication system based on two identical 4-D hyperchaotic oscillator systems.

2. Designing and derivation of two control approaches (Adaptive and LQR) for synchronization of the proposed hyperchaotic.

3. Presenting the numerical analyzes and MATLAB simulations to stabilize the chaotic system.

4. Realization of both synchronization methods using the FPGA.

5. Comparing the performance of two controllers in the synchronization of the proposed hyperchaotic system.

6. Generalizing comparison result by applying the synchronization on another family of chaotic systems using Adaptive control and LQR.

The remainder of the paper is structured as follows: The description of 4-D hyperchaotic systems is introduced in Section 2. Section 3 shows the sensitivity of a chaos-based secured communication system to the initial conditions and parameters variation. The synchronization of 4-D hyperchaotic systems using LQR and Adaptive control techniques is presented in section 4. Section 5 introduces the hardware realization on FPGA for the synchronization

systems using adaptive control and LQR. Comparing the performance indicators of the two controllers are presented in Section 6. Synchronization and performance indicator comparisons for a 3-D chaotic system is illustrated in section 7. Section 8 shows and discusses the results of the comparisons. We conclude this paper in Section 9.

## 2. 4-D hyperchaotic oscillator system

The Proposed 4-D system is derived from the Lorenz system, which is no-equilibrium hyperchaotic system. The dynamic equations of this system are represented in (2.1).

$$
\begin{aligned}
\dot{x}_1 &= x_2 - x_1 + x_4 \\
\dot{x}_2 &= -a_1 x_1 x_3 \\
\dot{x}_3 &= x_1 x_2 - 1 \\
\dot{x}_4 &= -a_2 x_1
\end{aligned}
\tag{2.1}
$$

where $x_1$, $x_2$, $x_3$, and $x_4$ are state variables. $a_1$ and $a_2$ are positive parameters. The system exhibits a chaotic behavior when the parameters $a_1$ and $a_2$ have the values: $10 \leq a_1 \leq 26$ $0.1 \leq a_2 \leq 1$, while the maximum hyperchaos occurs at: $a_1 = 18.3$ and $a_2 = 0.994$. The Lyapunov exponents (LE) of the hyperchaotic system are obtained as: $(L1, L2, L3, L4) = (0.2525, 0.0428, 0, -1.2953)$.

Fig 1A–1F illustrate phase attractors of the 4-D hyperchaotic oscillator. The fourth-order Runge-Kutta integrator with a fixed step size equals to (0.001) is used to produce the simulation results. The parameter values are $a_1 = 18.3$ and $a_2 = 0.994$ and the initial conditions are $x(0) = [1, -1, 1, -1]^T$. Please see [48] for a more detailed study of the dynamics of the proposed system.

## 3. Sensitivity of the 4-D hyperchaotic system to initial conditions and parameters changes

In this section, the proposed hyperchaotic system-based secured communication system is building and testing. Results show the effectiveness of the initial conditions and system parameters variation on the recovered message.

Fig 2 shows the Matlab block diagram of chaotic masking based secure communication system, where the information message $m(t)$ is added to the signal $x_m(t)$ output of Master hyperchaotic system in the transmitter side producing the signal $x_t(t)$ that will be sent through the channel. On the receiver side, the output $x_s$ of the slave hyper-chaotic system is subtracted from the received signal to extract the recovered information message $m_r(t)$.

To simulate the proposed secured communication system, the parameter values of master and slave are selected as $a_{1m} = a_{1s} = 18.3$, and $a_{2m} = a_{2s} = 0.994$. The initial conditions of the master and slave are selected as $X_m(0) = X_s(0) = [1, -1, 1, -1]^T$.

Fig 3A illustrates the originally transmitted message $m(t) = 0.1sin(\pi t)$ and the recovered received message $m_r(t)$ when no changes occurred in the initial conditions or the oscillator parameters. It is clear that both messages are identical.

### 3.1 Changing initial conditions

Obtaining a chaos-based secured communication system requires full matching between chaotic systems in the transmitter and the receiver which is quite impossible in reality because of the sensitivity of the chaotic systems. If a slight change in one initial value of our chaotic system occurred in the receiver, this will produce a dramatic difference between the recovered message and the original one. Fig 3B shows a comparison between the recovered message (in red) and the original one (in blue) versus time after changing the initial values $x_m(0) = [1, -1,$

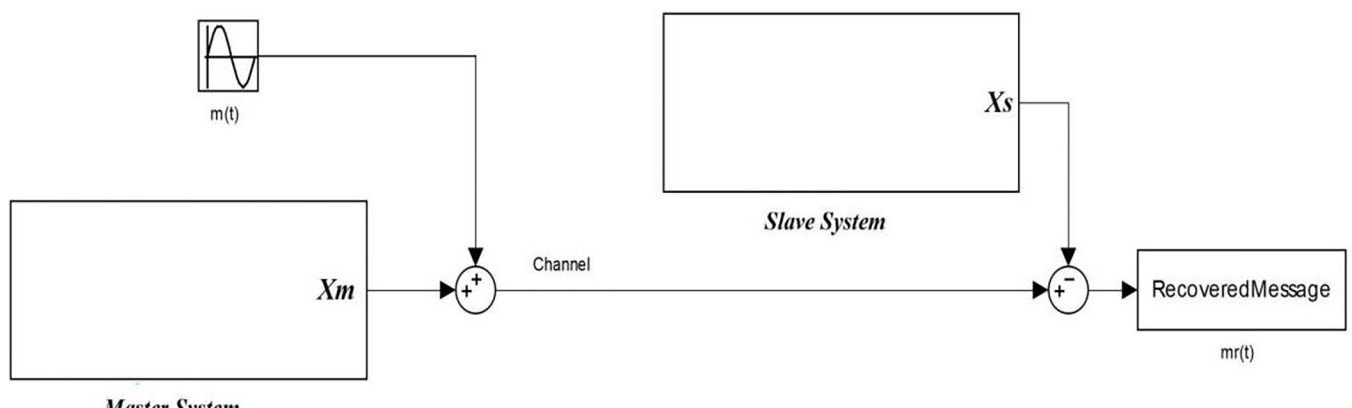

(a) $x_1 - x_2$　　(b) $x_1 - x_3$　　(c) $x_1 - x_4$

(d) $x_2 - x_3$　　(e) $x_2 - x_4$　　(f) $x_3 - x_4$

**Fig 1. Phase attractors of the 4-D hyperchaotic oscillator.** (a) $x_1 - x_2$, (b) $x_1 - x_3$, (c) $x_1 - x_4$, (d) $x_2 - x_3$, (e) $x_2 - x_4$, and (f) $x_3 - x_4$.

**Fig 2. Matlab/Simulink Block diagram of the proposed secured communication system.**

$1, -1]^T$ to $x_m(0) = [1, -1.0000001, 1, -1]^T$, where there are dramatic changes in the recovered message.

## 3.2 Changing system parameters

Fig 3C shows a comparison between the recovered message (in red) and the original one (in blue) versus time after changing the parameters value from [18.3, 0.994] to [18.3, 0.99]. This slight change destroys the recovered message clearly. To overcome these problems, the synchronization between chaotic systems is required.

# 4. Synchronization of 4-D hyperchaotic communication system

Since the identical chaotic systems with different initial conditions have different trajectories, so the synchronization between systems is required to minimize the error between the master and slave trajectories. This synchronization is implemented using two different control techniques for each oscillator in the slave side, Adaptive control and Linear Quadratic Regulator (LQR) based optimal control.

The process of applying the same controllers to systems with varying dimensions involves careful consideration of the underlying dynamics and control architecture. In our case, while the initial design was tailored for a 4-D system, the fundamental control principles remain applicable to systems with lower or higher dimensions.

## 4.1 Synchronization using adaptive control technique

This section includes deriving the Adaptive control law to synchronize the two identical proposed hyperchaotic systems. Considering the nonlinear 4-D hyperchaotic system described in (2.1), the response system with added control law can be written as follows:

$$
\begin{aligned}
\dot{y}_1 &= y_2 - y_1 + y_4 + u_1 \\
\dot{y}_2 &= -a_1 y_1 y_3 + u_2 \\
\dot{y}_3 &= y_1 y_2 - 1 + u_3 \\
\dot{y}_4 &= -a_2 y_1 + u_4
\end{aligned}
\tag{4.1}
$$

Such that $y_1$, $y_2$, $y_3$ and $y_4$ are state variables, $a_1$ and $a_2$ are positive parameters and $u = [u_1, u_2, u_3, u_4]$ is the control vector that keeps the slave system on the desired trajectory.

The dynamic error Eqs between (4.1) and (2.1) are obtained as following:

$$
\begin{aligned}
\dot{e}_1 &= \dot{y}_1 - \dot{x}_1 = e_2 - e_1 + e_4 + u_1 \\
\dot{e}_2 &= \dot{y}_2 - \dot{x}_2 = -a_1(y_1 y_3 - x_1 x_3) + u_2 \\
\dot{e}_3 &= \dot{y}_3 - \dot{x}_3 = (y_1 y_2 - x_1 x_2) + u_3 \\
\dot{e}_4 &= \dot{y}_4 - \dot{x}_4 = -a_2 e_1 + u_4
\end{aligned}
\tag{4.2}
$$

Then the Adaptive control law can be obtained as:

$$
\begin{aligned}
u_1 &= -e_2 + e_1 - e_4 - k_1 e_1 \\
u_2 &= A_1(y_1 y_3 - x_1 x_3) - k_2 e_2 \\
u_3 &= (y_1 y_2 - x_1 x_2) - k_3 e_3 \\
u_4 &= -A_2 e_1 - k_4 e_4
\end{aligned}
\tag{4.3}
$$

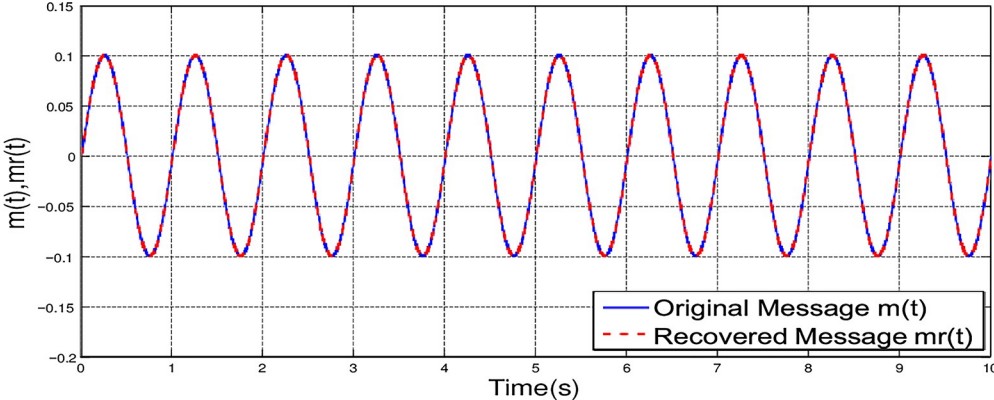

(a) $m(t)$ and $m_r(t)$ without any changes in the initial values or oscillator parameters

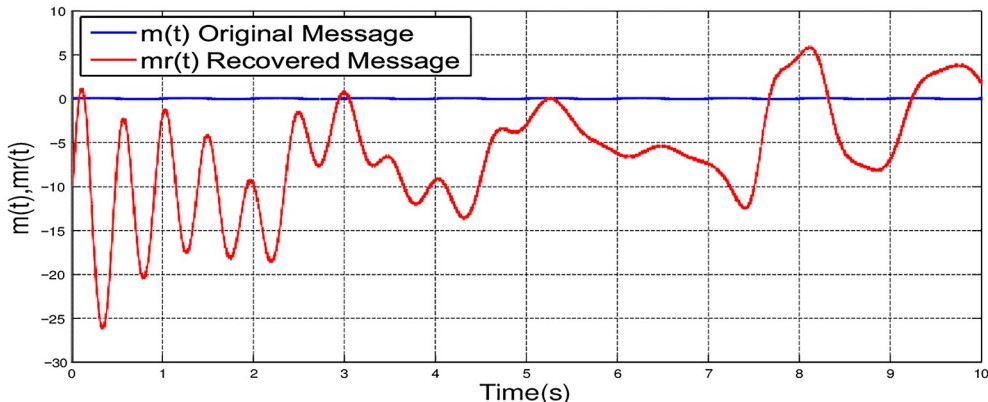

(b) $m(t)$ and $m_r(t)$ after changing one initial value in the receiver

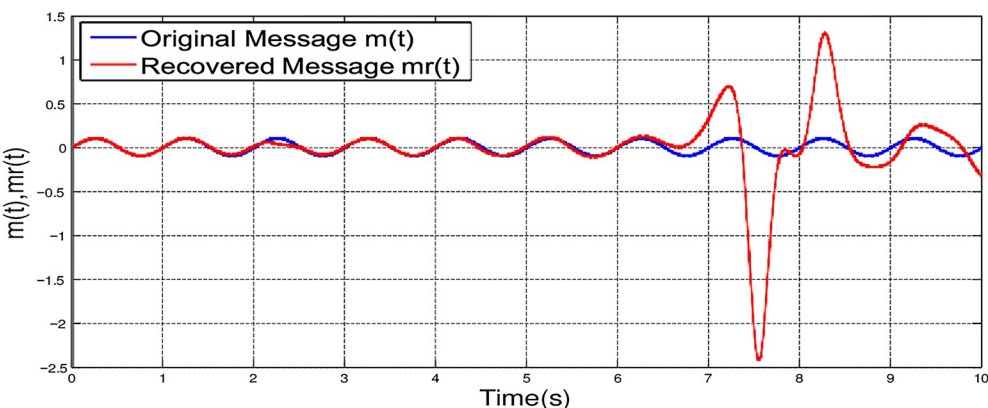

(c) $m(t)$ and $m_r(t)$ after changing one parameter value in the receiver

**Fig 3. The original message $m(t)$ and the recovered message $m_r(t)$.** (a) $m(t)$ and $m_r(t)$ without any changes in the initial values or oscillator parameters, (b) $m(t)$ and $m_r(t)$ after changing one initial value in the receiver and (c) $m(t)$ and $m_r(t)$ after changing one parameter value in the receiver.

Where $k_1$, $k_2$, $k_3$ and $k_4$ are positive gains and $A_1(t)$ and $A_2(t)$ are the unknown parameters estimation $a_1$ and $a_2$ respectively. The parameters estimation errors can be defined as given in (4.4):

$$e_{a1}(t) = a_1 - A_1$$
$$e_{a2}(t) = a_2 - A_2$$

(4.4)

By differentiation of (4.4) we obtained (4.5):

$$\dot{e}_{a1}(t) = -\dot{A}_1$$
$$\dot{e}_{a2}(t) = -\dot{A}_2$$

(4.5)

using (4.3) and (4.4), the dynamic error equations become:

$$\dot{e}_1 = -k_1 e_1$$
$$\dot{e}_2 = e_{a1}(y_1 y_3 - x_1 x_3) - k_2 e_2$$
$$\dot{e}_3 = -k_3 e_3$$
$$\dot{e}_4 = -e_{a2} e_1 - k_3 e_3$$

(4.6)

To find parameters estimate update laws, we use the Lyapunov stability theory. Let Lyapunov function defined as:

$$V = \frac{1}{2}(e_1^2 + e_2^2 + e_3^2 + e_4^2 + e_{a1}^2 + e_{b1}^2)$$

The first-order derivative of the Lyapunov function is:

$$\frac{dV}{dt} = (e_1 \dot{e}_1 + e_2 \dot{e}_2 + e_3 \dot{e}_3 + e_4 \dot{e}_4 + e_{a1} \dot{e}_{a1} + e_{b1} \dot{e}_{b1})$$

(4.7)

Substituting Eqs (4.5) and (4.6) in (4.7), we get:

$$\frac{dV}{dt} = (-k_1 e_1^2 - k_2 e_2^2 - k_3 e_3^2 - k_4 e_4^2 + e_{a1}(-e_2(y_1 y_3 - x_1 x_3) + \dot{A}_1) + e_b(-e_1 e_4 + \dot{A}_2)$$

(4.8)

From the previous equation, we get the parameters update law as:

$$\dot{A}_1 = e_2(y_1 y_3 - x_1 x_3)$$
$$\dot{A}_2 = -e_1 e_4$$

(4.9)

Substituting the parameter update law in (4.9) into (4.8), we obtain:

$$\frac{dV}{dt} = (-k_1 e_1^2 - k_2 e_2^2 - k_3 e_3^2 - k_4 e_4^2)$$

The output tracking error can then asymptotically converge to zero according to Lyapunov stability theory in the sense that $\frac{dv}{dt} < 0$ and $k_1$, $k_2$, $k_3$, and $k_4$ are positive constants. So according to the Lyapunov stability theory [49], the error dynamics in (4.6) is globally asymptotically stable.

The simulation block diagram based on these equations is shown in Fig 4. It is represented using four blocks; master chaotic system, parameter estimation, adaptive controller, and slave chaotic system blocks. X and Y indicate state vectors of master and slave systems. Respectively, while u is a vector of the Adaptive control law.

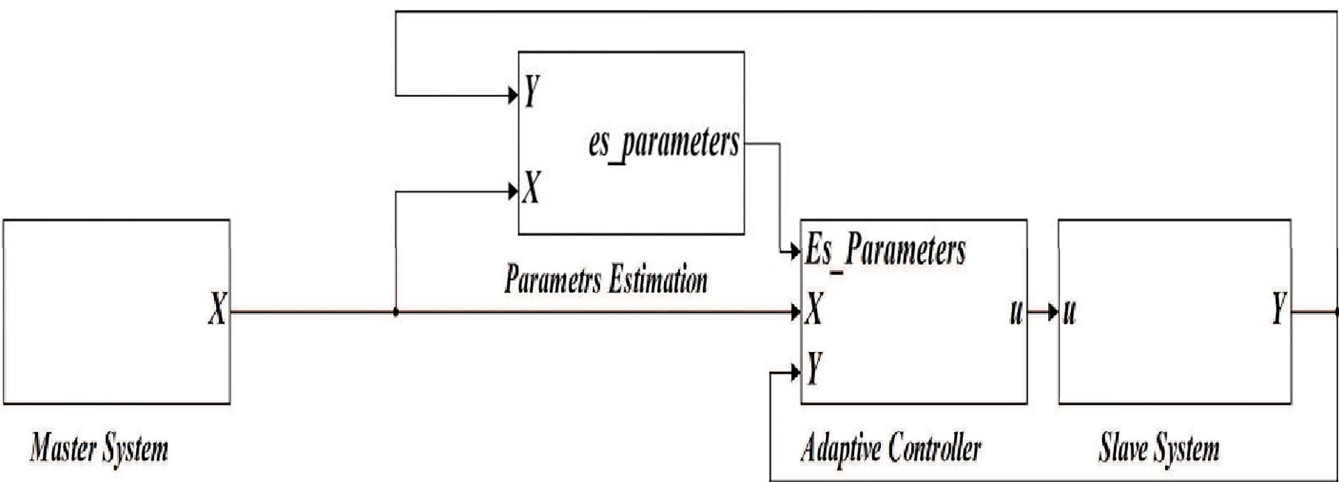

**Fig 4. Matlab/Simulink block diagram of the synchronized 4-D hyperchaotic system using adaptive control.**

Based on the previous equations, the detailed Matlab/Simulink representation of these blocks are shown in Figs 5–8.

Fig 5 illustrates Matlab/Simulink representation of the master blocks of the 4- D hyperchaotic system given in Eq (2.1). The master system consists of four main parts, one for each state variable where $x(1)$, $x(2)$, $x(3)$, $x(4)$ represents the state variable vector of the master system. Similarly, Fig 6 shows the Matlab/Simulink representation of the slave blocks of the 4-D hyperchaotic system given in Eq (4.1).

In this Fig 6, $y(1)$, $y(2)$, $y(3)$, $y(4)$ represent the state variable vector of the slave system, while $u(1)$, $u(2)$, $u(3)$, $u(4)$ represent the control vector that keeps the slave system on the desired trajectory.

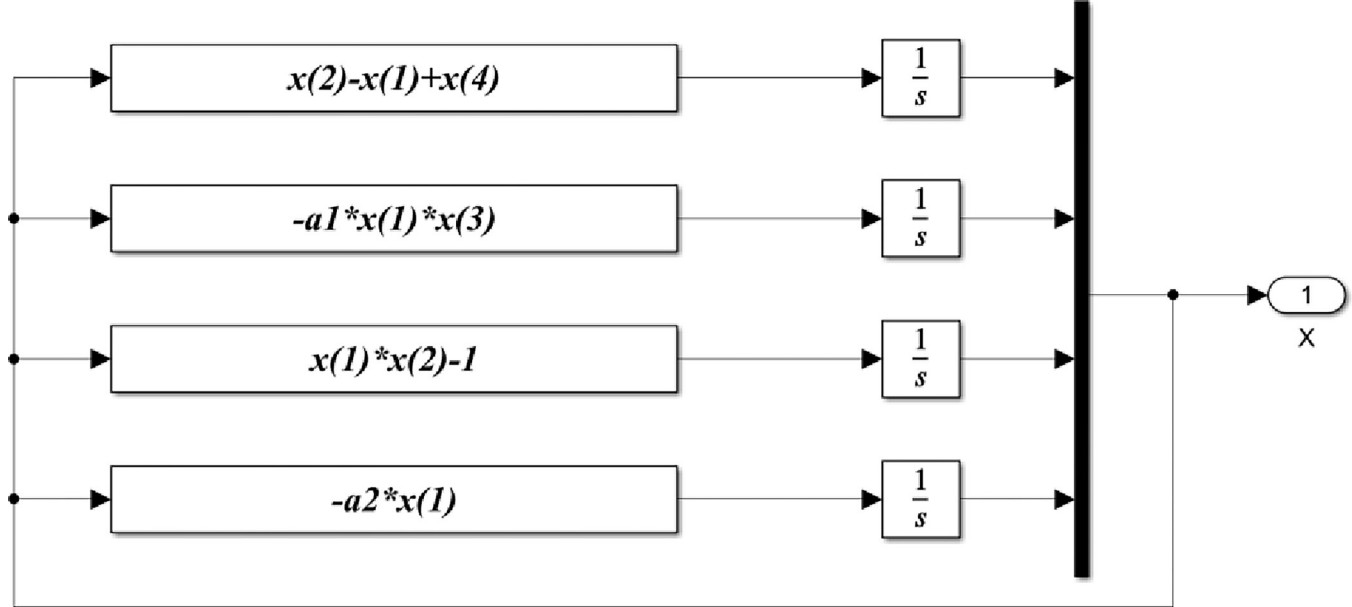

**Fig 5. Matlab/Simulink representation of the master block of the 4-D hyperchaotic system.**

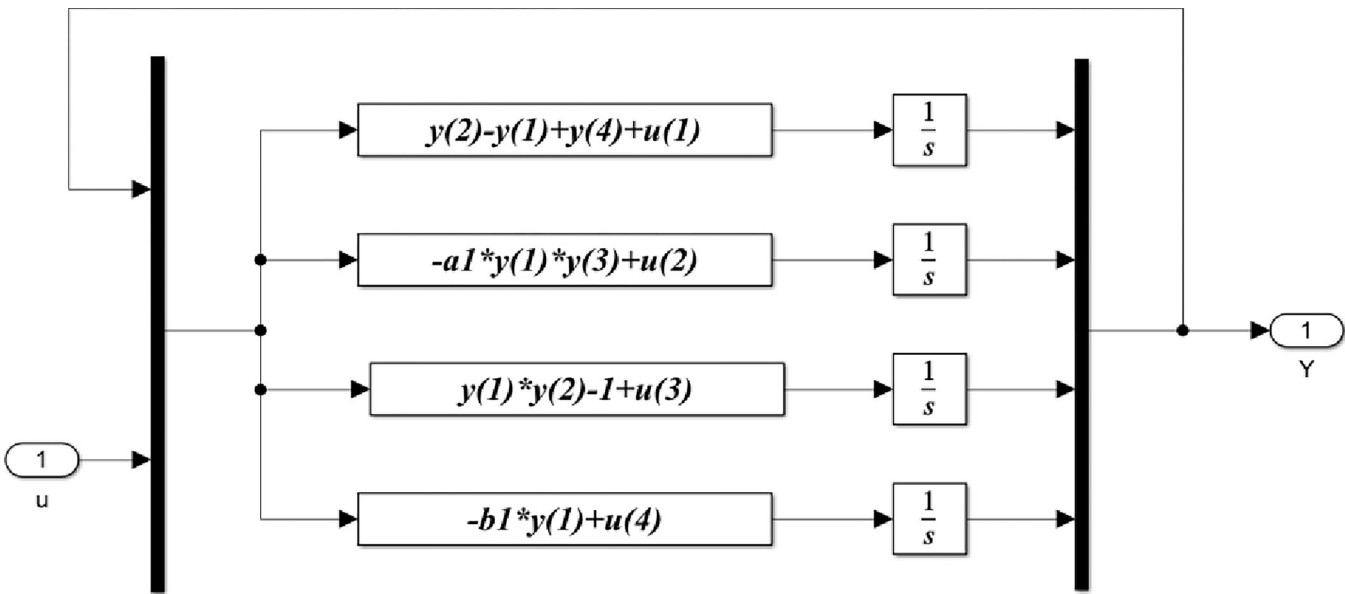

**Fig 6. Matlab/Simulink representation of the slave block of the 4-D hyperchaotic system.**

The Matlab/Simulink representation of the parameter estimation block shown in Fig 7 indicates the integration of parameter update laws in (4.9) based on the error between the slave and the master systems. Where in Fig 7, $x(1)$, $x(2)$, $x(3)$, $x(4)$ and, $y(1)$, $y(2)$, $y(3)$, $y(4)$ represent state variable vectors of the master and slave system respectively.

The representation of the Adaptive controller block is shown in Fig 8. The block indicates the derived Adaptive control laws based on (4.3) and the values of parameters from the estimated parameters block.

## 4.2 Synchronization using linear quadratic regulator (LQR)

The objective of the optimal control is to determine the control vector u(t) that is added to the dynamic equations on the slave side. This control forces the behavior of the system under control (slave oscillator) to minimize the cost function and to maximize the return from the

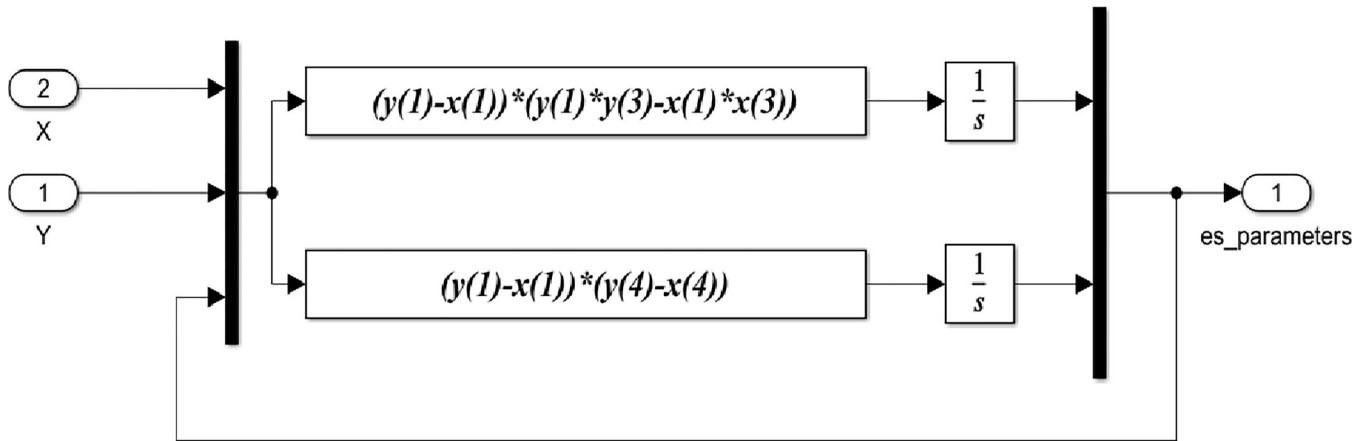

**Fig 7. Matlab/Simulink representation of the parameter estimation block of the 4-D chaotic system.**

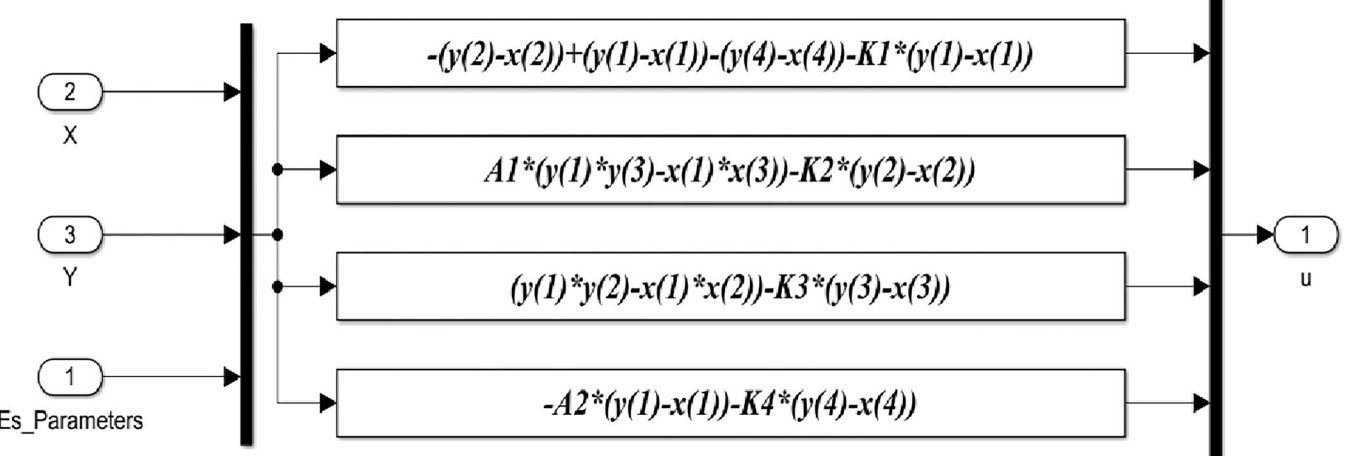

**Fig 8. Matlab/Simulink representation of the adaptive controller block of the 4-D hyperchaotic system.**

system at the same time. The optimal control methodology seeks to control a system in the most preferred form, taking into account a cost index that includes optimization metrics.

For the derivation of the linear quadratic regulator, we consider a state-space representation for the non-linear master and slave systems,

$$\dot{x} = Ax + g(x)$$
$$\dot{y} = Ay + g(y) + u$$

Where $x, y \in R^n$ are state vectors; $A \in R^{n \times n}$ is the matrix of the linear terms, g(.) is the vector of the continuous non-linear functions and $u \in R^m$ is a control vector that stabilizes the slave system in the desired trajectory. The error vector of the system defined as:

$$e = y - x$$
$$e = A\dot{e} + h(x, y) + u \qquad (4.10)$$

Where $h(x, y) = g(y) - g(x)$. The aim of control law $u$ is to satisfy $\lim(e \rightarrow 0)$ as $t \rightarrow \infty$. If u is the sum of two linear and non-linear terms as below;

$$u = -h(x, y) + Bu_l \qquad (4.11)$$

Where $B \in R^{n \times m}$, then substituting (4.11) into (4.10)

$$\dot{e} = Ae + Bu_l$$

consider the linear control term is given by:

$$u_l = -Ke$$

Where $K \in R^{m \times n}$ is the linear gain matrix, then the dynamic error can be written as:

$$\dot{e} = (A - BK)e$$

Here, the gain K can be obtained using a linear control method (Linear Quadratic Regulator LQR). The consider the cost function as follows:

$$J = \int_0^\infty f(x, u) = \int_0^\infty [X^T Q X + u_l^T R u_l] \tag{4.12}$$

Where Q and R matrices are positive definite matrices to ensure the cost function remains positive. From optimal control theory [50], the gain of optimal control is given by:

$$u_l = -R^{-1} B^T P e$$

Where P is the positive symmetrical matrix that gives the solution for Algebraic Ri-catty Equation (ARE)

$$0 = Q + A^T P + PA - PBR^{-1}B^T P$$

For the 4-D hyperchaotic system, we choose $Q = 100.I_{4\times4} R = I_{4\times4}$. Now, the control law could be achieved by solving the Ricatti equation in MATLAB using the 'lqr' command to determine P and K matrices.

$$P = K = \begin{bmatrix} 9.0442 & 0.4494 & 0.0000 & 0.0449 \\ 0.4494 & 10.0348 & 0.0000 & 0.0212 \\ 0.0000 & 0.0000 & 10.0000 & 0.0000 \\ 0.0449 & 0.0212 & 0.0000 & 9.9954 \end{bmatrix}$$

The simulation block diagram using LQR is shown in Fig 9. It is represented using three blocks; master chaotic system, LQR controller, and slave chaotic system blocks. X and Y indicate state vectors of master and slave systems. Respectively, while u is a vector of LQR control law. The master and slave chaotic system blocks of the 4-D hyperchaotic oscillator are built-in Matlab/Simulink as shown in Figs 5 and 6, respectively.

The representation of the LQR block is shown in Fig 10. The block indicates the derived control law based on (4.11) and the gain matrix K of the LQR controller.

## 5. FPGA hardware realization

In this section, we show how to realize the complete 4-D hyperchaotic system including the two controllers (Adaptive and LQR) on FPGA platform. First, the numerical solution of the 4-D system are obtained using forward Euler integration method. Then, it is realized on the FPGA platform using VHDL code. For the implementation on the FPGA, we use the 32-bits

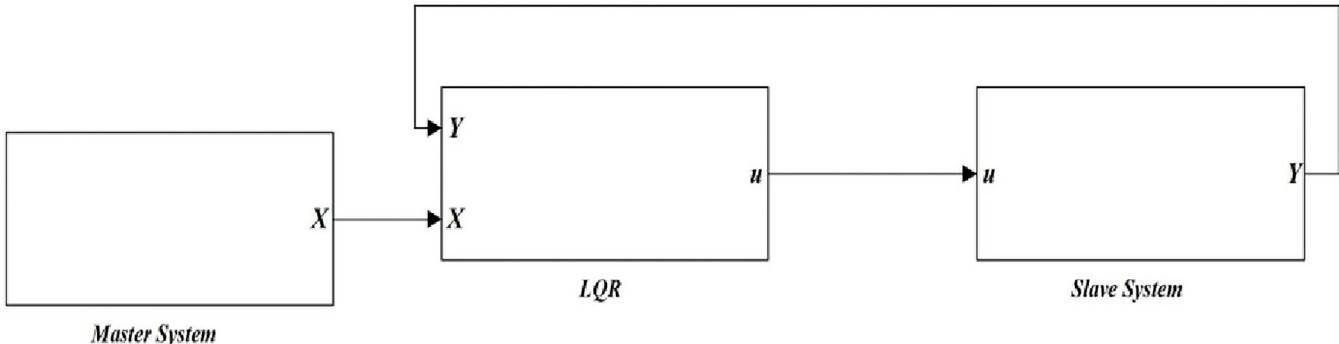

**Fig 9. Matlab/Simulink block diagram of the synchronized 4-D hyperchaotic system using LQR.**

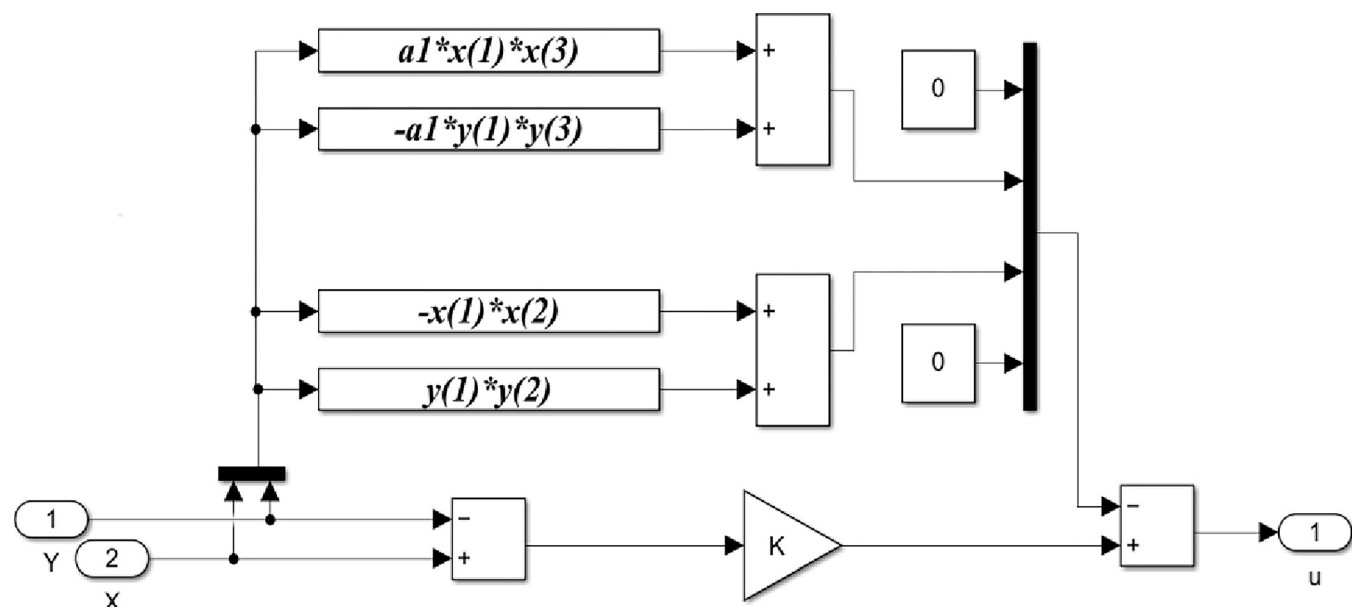

**Fig 10. Matlab/Simulink detailed representation of the LQR block of the 4-D hyper-chaotic system.**

fixed-point arithmetic representation for all the variables and constants. This representation has 1-bit for the sign, 7-bits for the integer part, and 24- bits for the fractional part. The basic hardware blocks required for the implementation are adders, subtractors and multipliers.

Fig 11 shows the top-level block diagram for the synchronization of 4-D system. The block diagram consists of three basic blocks, the master system where the output of this system is [$x_1$, $x_2$, $x_3$, $x_4$], the controller which compares between the master and slave system to generate the control vector [$u_1$, $u_2$, $u_3$, $u_4$], and the slave system which receive the control vector to synchronous the masetr system. where the output vector of the slave system is [$y_1$, $y_2$, $y_3$, $y_4$]. In the next subsections the hardware realization of each block is described.

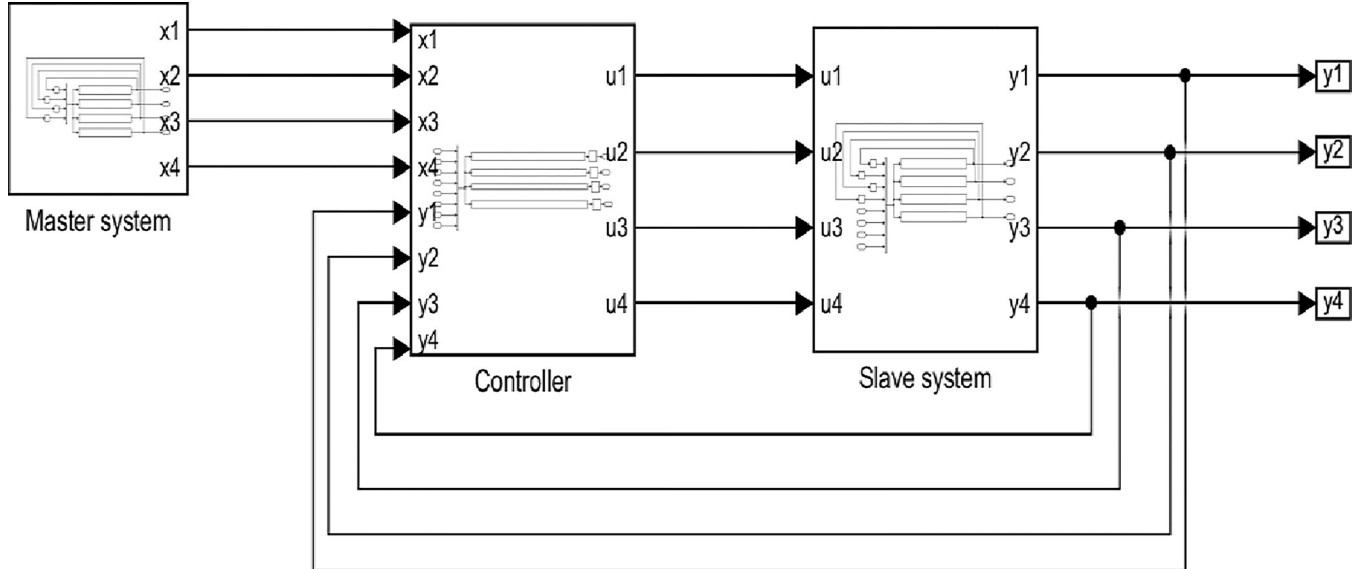

**Fig 11. Top level block diagram for the synchronization of 4-D hyperchaotic system.**

## 5.1 Realization of the master system

Euler's equations of the 4-D master hyperchaotic system are shown in (5.1):

$$
\begin{aligned}
x_1[n+1] &= x_1[n] + h(x_2[n] - x_1([n] + x_4[n]) \\
x_2[n+1] &= x_2[n] + h(-a_1 x_1[n] x_3[n]) \\
x_3[n+1] &= x_3[n] + h(x_1[n] x_2[n] - 1) \\
x_4[n+1] &= x_4[n] + h(-a_2 x_1[n])
\end{aligned}
\tag{5.1}
$$

where n and n + 1 are the current and the next states, respectively. $x_1$, $x_2$, $x_3$, $x_4$ are output state variables of the master system, $a_1 = 18.3$ and $a_2 = 0.994$ are the coefficient parameters. The initial conditions of the state variables are $x_0 = [1, -1, 1, -1]$, while the discretization step size is h=0.001.

Fig 12 shows the Basic block connections to implement the Eq (5.1) of the master system on FPGA. The implementation requires 5 adders, 2 subtractors, and 6 multipliers.

## 5.2 Realization of the slave system

Euler's equations of the 4-D slave hyperchaotic system are shown in (5.2):

$$
\begin{aligned}
y_1[n+1] &= y_1[n] + h(y_2[n] - y_1[[n] + y_4[n]) + u_1 \\
y_2[n+1] &= y_2[n] + h(-a_1 y_1[n] y_3[n]) + u_2 \\
y_3[n+1] &= y_3[n] + h(y_1[n] y_2[n] - 1]) + u_3 \\
y_4[n+1] &= y_4[n] + h(-a_2 y_1[n]) + u_4
\end{aligned}
\tag{5.2}
$$

where $y_1$, $y_2$, $y_3$, $y_4$ are the output state variables of the slave system, while $u_1$, $u_2$, $u_3$, $u_4$ are the input control laws of the slave system. The initial conditions of the slave system are $y_0 = [-1, 1, -1, 1]$, and the discretization step size is h=0.001.

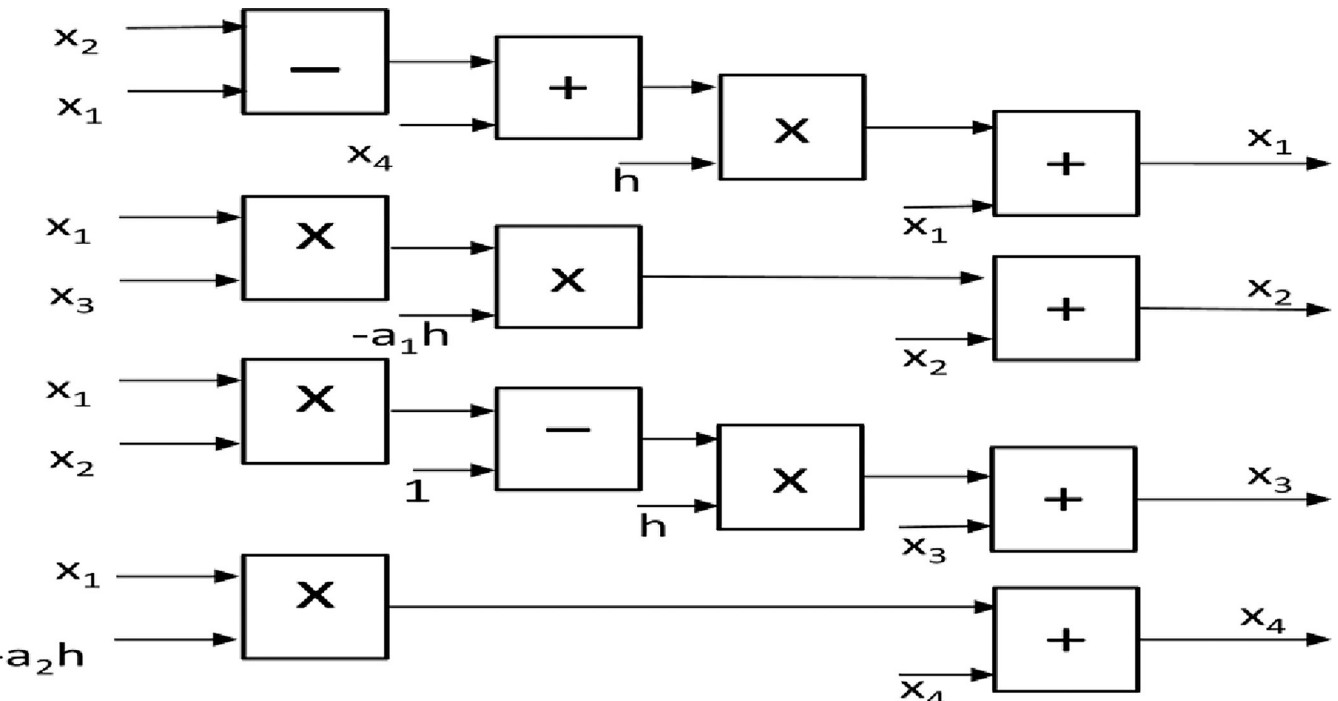

**Fig 12. Basic blocks connections required to implement the master system on FPGA.**

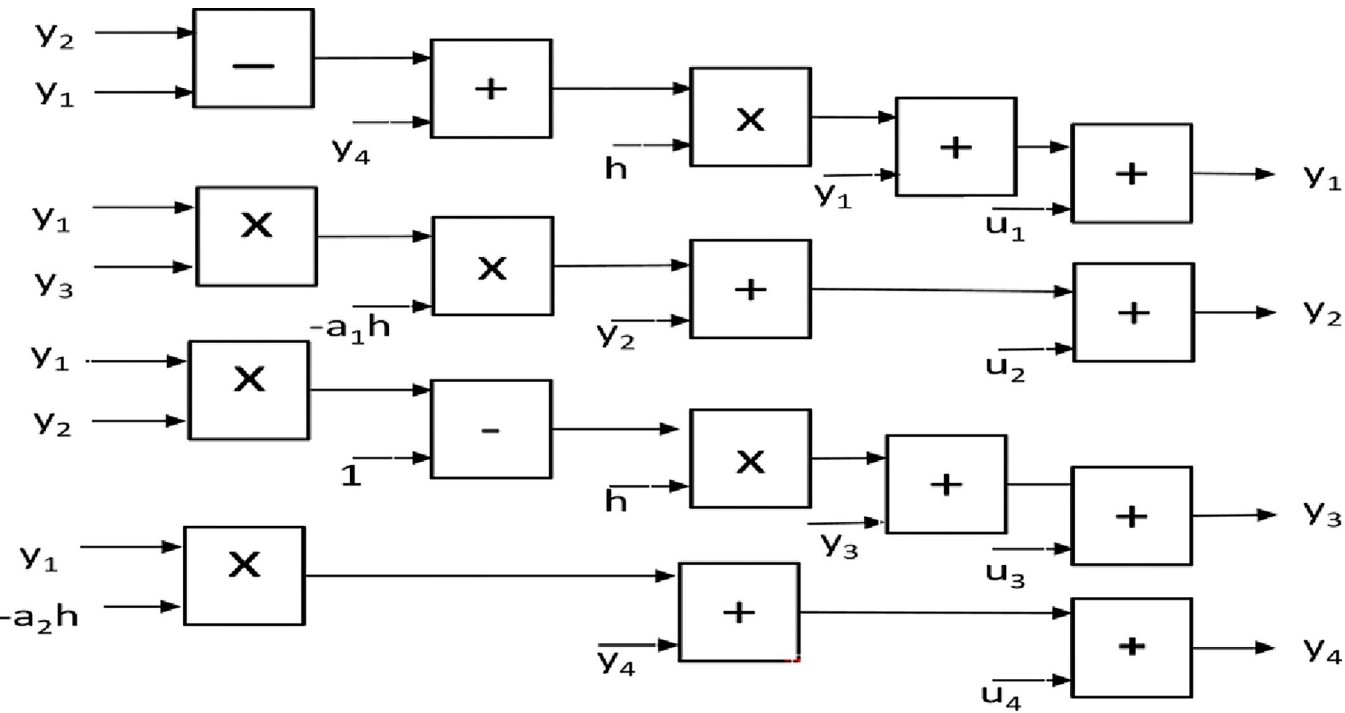

**Fig 13. Basic blocks connections required to implement the slave system on FPGA.**

Fig 13 shows the Basic block connections to implement the Eq (5.2) of the slave system on FPGA. Here, the implementation requires 9 adders, 2 subtractors, and 6 multipliers.

## 5.3 Realization of the adaptive controller

The controller block in Fig 11 depends on the synchronization method used in the communication system. The numerical solution of the adaptive control using forward Euler method is shown in (5.3)

$$
\begin{aligned}
u_1 &= -(y_2[n] - x_2[n]) + (y_1[n] - x_1[n]) - (y_4[n] - x_4[n]) - K_1(y_1[n] - x_1[n]) \\
u_2 &= A_1[n](y_1[n]y_3[n] - x_1[n]x_3[n]) - K_2(y_2[n] - x_2[n]) \\
u_3 &= (y_1[n]y_2[n] - x_1[n]x_2[n]) - K_3(y_3[n] - x_3[n]) \\
u_4 &= -A_2[n](y_1[n] - x_1[n]) - K_4(y_4[n] - x_4[n])
\end{aligned}
\tag{5.3}
$$

Where $u_1$, $u_2$, $u_3$, $u_4$ are control laws, $K_1$, $K_2$, $K_3$, $K_4$ are the adaptive controller gains, and $A_1$, $A_2$ are the estimated parameters which are given as:

$$
\begin{aligned}
A_1[n+1] &= A_1[n] + h((y_2[n] - x_2[n])(y_1[n]y_3[n] - x_1[n]x_3[n])) \\
A_2[n+1] &= A_2[n] + h((y_1[n] - x_1[n])(y_4[n] - x_4[n]))
\end{aligned}
\tag{5.4}
$$

Fig 14 shows the Basic block connections to implement the Eq (5.3) of the adaptive controller on FPGA. It is worth noting that the implementation requires more resources than the other blocks. Here, the resources are 15 subtractors, and 10 multipliers.

As shown in Fig 14, some blocks have the estimated parameters $A_1$, $A_2$ as inputs.

Fig 15 shows the Basic block connections of these parameters as defined in Eq (5.4).

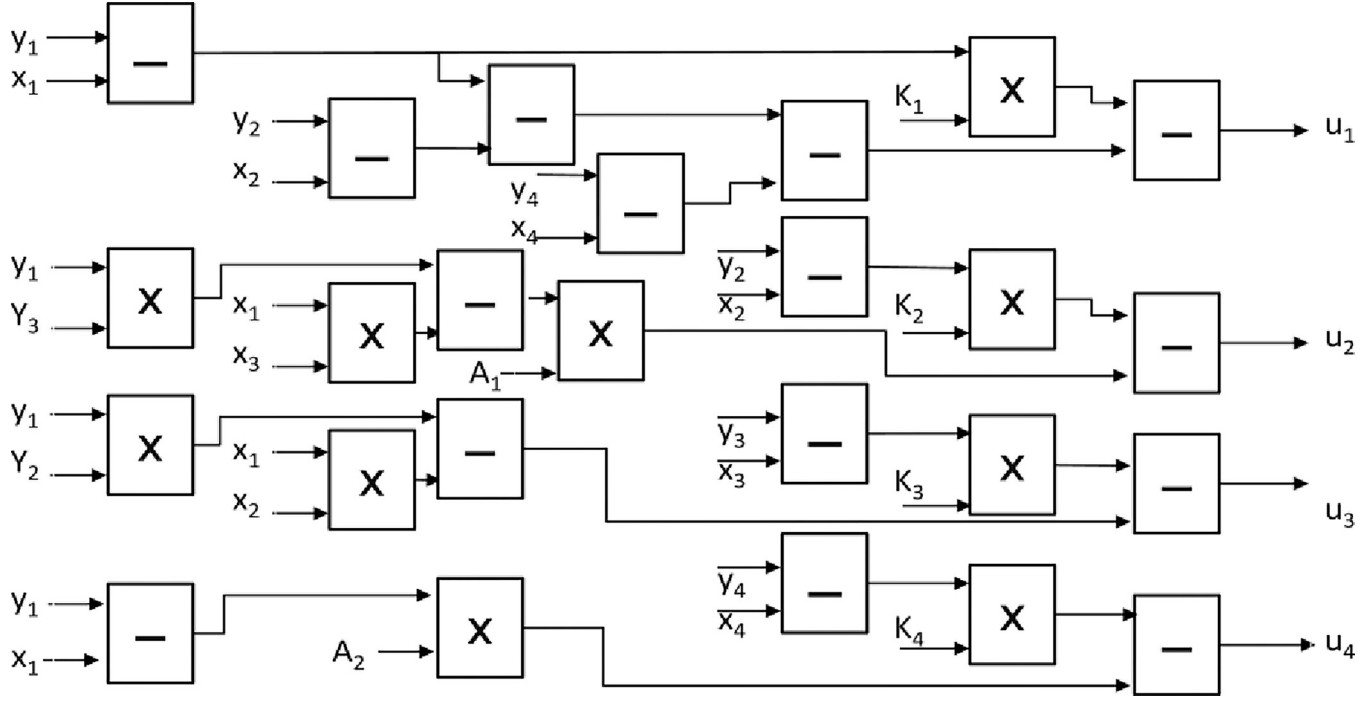

**Fig 14. Basic blocks connections required to implement the adaptive controller on FPGA.**

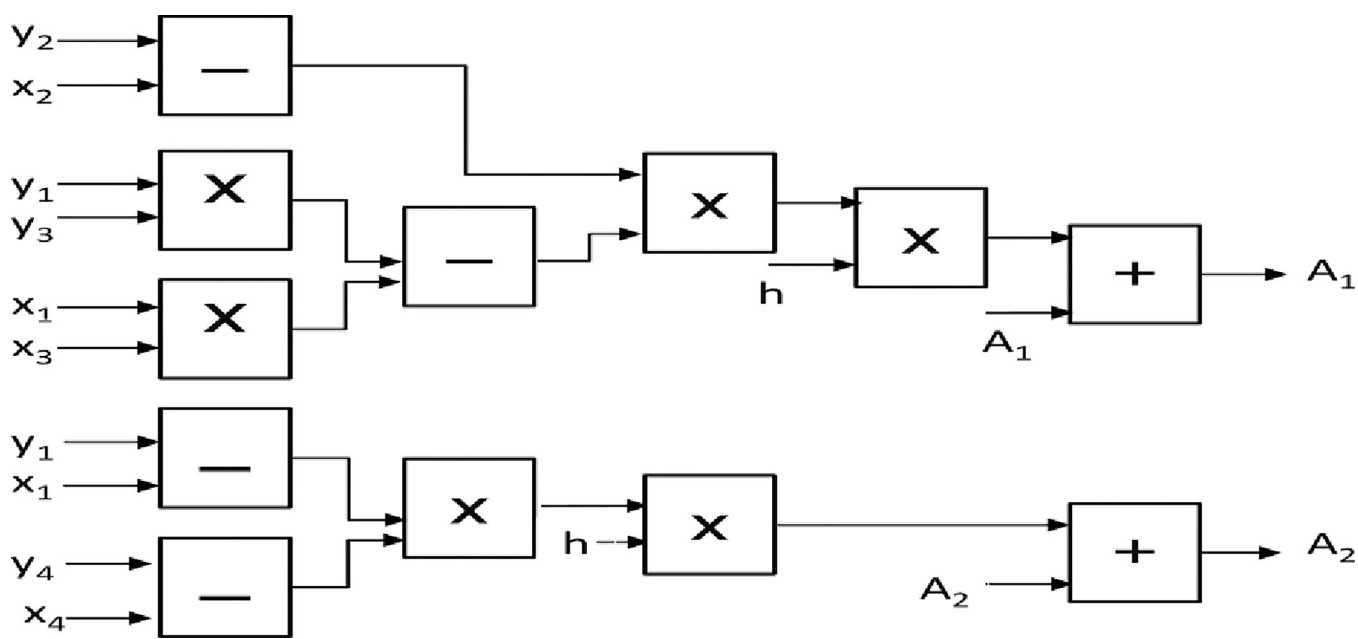

**Fig 15. Basic blocks connections required to implement the parameter estimation laws of the adaptive controller on FPGA.**

### 5.4 Realization of the LQR controller

The numerical solution of the LQR controller is given in (5.5)

$$u_1 = K_{11}(x_1[n] - y_1[n]) + K_{12}(x_2[n] - y_2[n]) + K_{13}(x_3[n] - y_3[n]) + K_{14}(x_4[n] - y_4[n])$$
$$u_2 = K_{21}(x_1[n] - y_1[n]) + K_{22}(x_2[n] - y_2[n]) + K_{23}(x_3[n] - y_3[n]) + K_{24}(x_4[n] - y_4[n])$$
$$u_3 = K_{31}(x_1[n] - y_1[n]) + K_{32}(x_2[n] - y_2[n]) + K_{33}(x_3[n] - y_3[n]) + K_{34}(x_4[n] - y_4[n]) \quad (5.5)$$
$$u_4 = K_{41}(x_1[n] - y_1[n]) + K_{42}(x_2[n] - y_2[n]) + K_{43}(x_3[n] - y_3[n]) + K_{44}(x_4[n] - y_4[n])$$

where $K_{11}$ till $K_{44}$ are the LQR control gains

Fig 16 shows the Basic block connections to implement LQR system on FPGA (only $u_1$ is shown for brevity). The total number of basic blocks required for implementation is 12 adders, 16 substractors, and 16 multipliers.

## 6. Comparing the performance indicators of the two controllers

Fig 17A–17D illustrate the state space synchronization using two control methods, while Fig 17E shows a comparison of the synchronization error for the two methods. Where the reference signals are marked in blue color, the responses using LQR and adaptive control are marked with green and red colors respectively. Each signal starts from its initial value till the matching between the master and slave trajectories is achieved by the LQR.

In this section, a comparison analysis is performed based on some performance indicators such as the error signal, the energy applied from the controller, response time, robustness to variation, etc. The performance indicators are compared based on two cases; initial conditions variation and parameters uncertainty.

### 6.1 Variation in initial conditions

Since chaotic systems are very sensitive to initial conditions, so, this will be a good indicator to measure the performance of proposed controllers. In this section, we change the initial

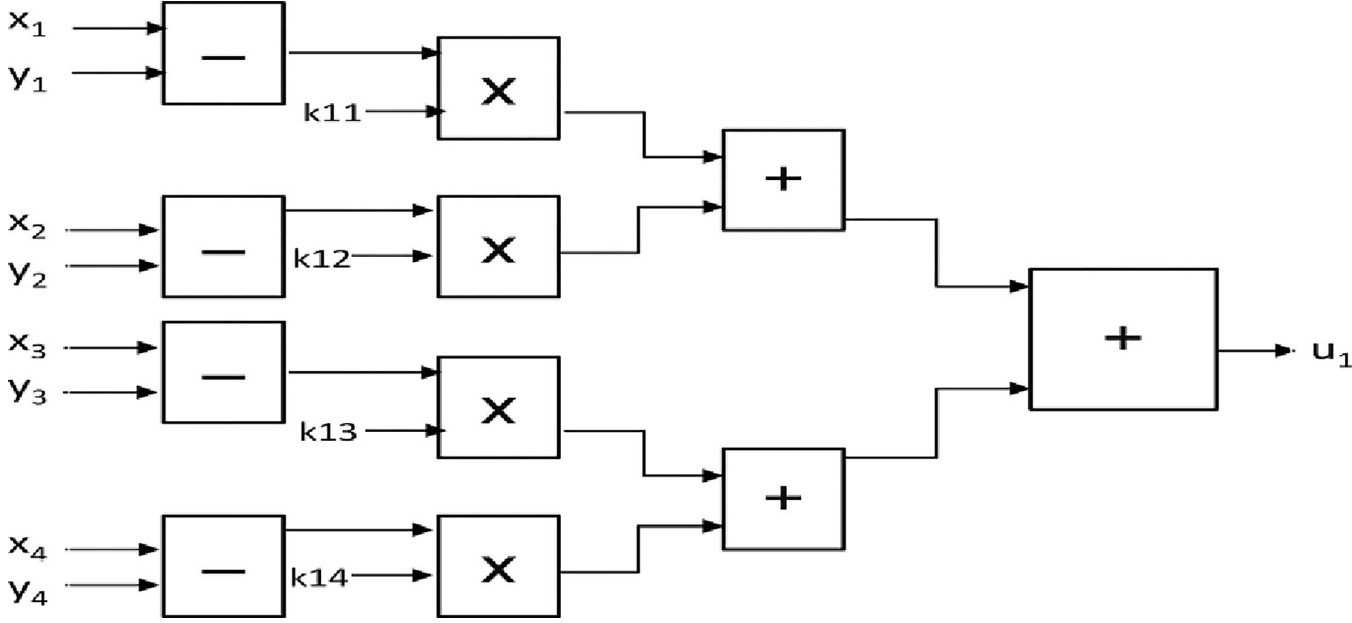

**Fig 16. Basic blocks required to implement the LQR system on FPGA.**

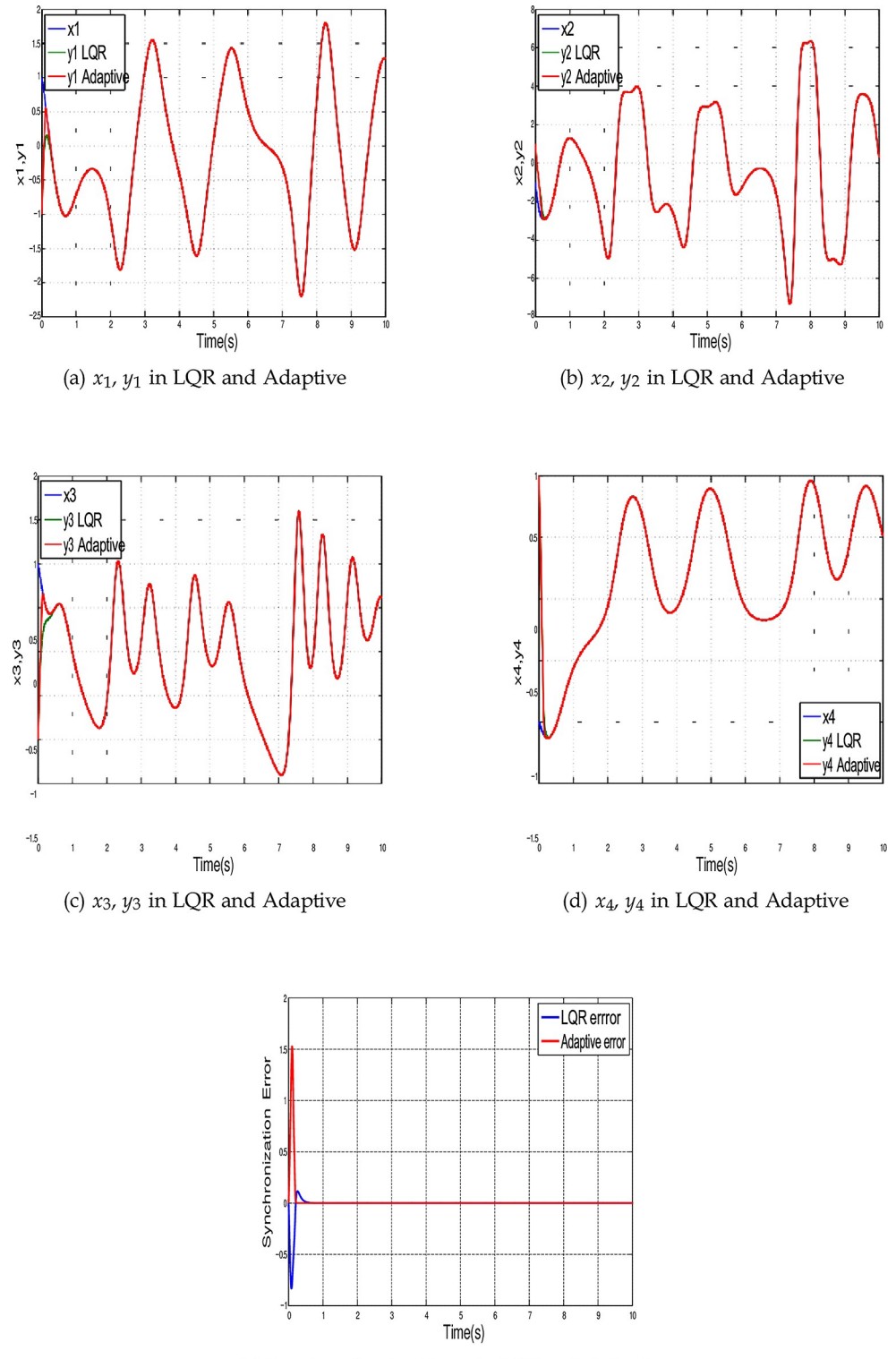

(a) $x_1$, $y_1$ in LQR and Adaptive

(b) $x_2$, $y_2$ in LQR and Adaptive

(c) $x_3$, $y_3$ in LQR and Adaptive

(d) $x_4$, $y_4$ in LQR and Adaptive

(e) Synchronization Error in LQR and Adaptive

**Fig 17. Time history of the synchronized 4-D hyperchaotic system using LQR and Adaptive control methods.** (a) $x_1$, $y_1$ in LQR and Adaptive, (b) $x_2$, $y_2$ in LQR and Adaptive, (c) $x_3$, $y_3$ in LQR and Adaptive, (d) $x_4$, $y_4$ in LQR and Adaptive, and (e) Synchronization Error in LQR and Adaptive.

conditions and compare the following performance indicators of the two control methods used for the proposed system:

1. Response Time ($t_r$):
   The response time for each controller is the time required for the state space of the slave system to synchronize the master system. In other words, it is a measure of the controller's response speed.

2. Integral Square Error (ISE):
   Integral Square Error is a performance indicator that is used in control theory to compare control methods, obtained by integration of the system's error over a fixed time interval.

$$ISE = \sum_{i=1}^{4} \int_{0}^{\infty} (e_i)^2 dt \qquad (6.1)$$

Where $e_i = [e_1, e_2, e_3, e_4]$ is the error vector

3. Control Effort (Ue):
   The control effort is the amount of energy or power necessary for the controller to perform its duty.

$$Ue = \sum_{i=1}^{4} \int_{0}^{\infty} (u_i)^2 dt$$

Where $u_i = [u_1, u_2, u_3, u_4]$ is the control vector applied from controllers.

4. Cost function (J):
   It is a function that measures the performance of the control technique in which the return value of this function must be as small as possible. The form of cost index used in this comparison is given in (6.2) which is a combination of the two previous criteria ISE and Eu.

$$J = \int_{0}^{\infty} e^T Q e + u^T R u \; dt \qquad (6.2)$$

where R and Q are weight matrices defined as $R = I_{4\times4}$ and $Q = 10I_{4\times4}$.
Table 1 illustrates the values of performance indicators mentioned above for both control

**Table 1. The performance indicators for the synchronized 4-D hyperchaotic system using adaptive control and LQR for different values of initial conditions.**

| Case | Initial Condidtions | | | | Response Time tr (s) | | ISE | | Ue | | J | |
|---|---|---|---|---|---|---|---|---|---|---|---|---|
| Number | Y1(0) | Y2(0) | Y3(0) | Y4(0) | LQR | Adaptive Control | LQR | Adaptive Control | LQR | Adaptive Control | LQR | Adaptive Control |
| 1 | -4 | -3 | -2 | -1 | 1.07 | 0.695 | 4.6085 | 4.4073 | 154.0615 | 158.4443 | 200.1465 | 202.5173 |
| 2 | -3.5 | -2.5 | -1.5 | -0.5 | 1.354 | 0.635 | 2.5205 | 2.4580 | 123.5567 | 147.4859 | 148.7622 | 172.0661 |
| 3 | -3 | -2 | -1 | 0 | 0.998 | 0.59 | 1.4725 | 1.4622 | 106.6579 | 141.2941 | 121.3831 | 155.9162 |
| 4 | -2.5 | -1.5 | -0.5 | 0.5 | 1.25 | 0.566 | 0.9963 | 0.9930 | 96.6740 | 130.6932 | 106.6371 | 140.6230 |
| 5 | -2 | -1 | 0 | 1 | 1.17 | 0.535 | 0.7794 | 0.8018 | 99.0182 | 122.4532 | 106.8123 | 130.4712 |
| 6 | -1.5 | -0.5 | 0.5 | 1.5 | 0.87 | 0.54 | 0.8491 | 0.8544 | 100.9536 | 115.5450 | 109.4446 | 124.0891 |
| 7 | -1 | 0 | 1 | 2 | 0.87 | 0.59 | 1.0755 | 1.0052 | 99.4098 | 106.2487 | 110.1646 | 116.3011 |
| 8 | -0.5 | 0.5 | 1.5 | 2.5 | 0.87 | 0.54 | 1.3781 | 1.3003 | 95.1868 | 111.2408 | 108.9681 | 124.2441 |
| 9 | 0 | 1 | 2 | 3 | 1.24 | 0.629 | 1.7927 | 1.7139 | 91.1529 | 115.7136 | 109.0802 | 132.8530 |
| 10 | 0.5 | 1.5 | 2.5 | 3.5 | 1.294 | 0.648 | 2.3792 | 2.3049 | 95.2507 | 115.4982 | 119.0431 | 138.5472 |
| Average | | | | | 1.1695 | 0.5968 | 1.7852 | 1.7301 | 106.1922 | 126.4617 | 124.0442 | 143.7628 |
| Differance in percent | | | | | 48.97% | | 3.09% | | 16.03% | | 13.72% | |

methods. It shows a comparison between Adaptive control and LQR in the synchronization of two identical 4-D hyperchaotic systems for ten different initial conditions of the slave system (from case number 1 till case number 10).

The first column indicates case number related to the initial values in the columns (2 to 5) corresponding to each state variable, while the rest of the columns indicate performance indicators related to LQR and Adaptive control. The last two rows of Table 1 refer to the average values for each column and the percentage of the difference between the two studied techniques for each performance indicator, respectively.

The graphical representations of Table 1 are shown in Fig 18A–18D that related to the performance indicators $t_r$, ISE, Ue and J, respectively, using both control techniques LQR (blue color) and Adaptive control (red color) for different values of initial conditions.

Fig 18A indicates the response time $t_r$ for both controllers using ten different initial conditions, the speed response of the Adaptive control is higher than the LQR for all random values of initial conditions. The minimum value of $t_r$ for the LQR is 0.87 s (at case number 7, where the initial conditions are $y(0) = [-1, 0, 1, 2]^T$). For the Adaptive control the minimum time is 0.535 s (at case number 5 where the initial conditions are $y(0) = [-2, -1, 0, 1]^T$). While the average values of response time for all ten different cases are 1.1695 s and 0.5968 s for LQR and Adaptive control respectively. This shows that the speed response of Adaptive control is 48.97% higher than the LQR.

The ISE for both control methods is shown in Fig 18B. The values of the ISE are almost the same for both controllers through all different initial conditions. The average values of ISE are 1.7852 and 1.7301 for LQR and Adaptive control respectively.

Fig 18C shows the control effort for each controller through different initial conditions. As shown in this figure, the LQR requires energy less than the Adaptive control for all proposed initial conditions. The minimum value for the energy applied from LQR is 91.1529 (at case number 9). For the Adaptive control, the minimum energy is 106.2487 (at case number 7). The average values of energy are 106.1922 and 126.4617 for LQR and Adaptive control respectively. This shows that energy consumption by LQR is 16.03% more than Adaptive control.

Fig 18D indicates the values of cost function for both control methods. As shown in this figure, the LQR requires less cost than Adaptive control. The minimum value for the cost indicator related to the LQR is 106.6371 (at case number 4). For the Adaptive control the minimum cost is 116.3011 (at case number 7). The average values of cost function are 124.0442 and 143.7628 for LQR and Adaptive control respectively. As expected, the returns of cost function for LQR is better than Adaptive control by 13.72% since it's designed based on this purpose.

## 6.2 Parameter uncertainty

In the case of parameter uncertainty, the robustness of each control technique is considered by adding uncertainty to the parameters in the slave system. Each control technique is designed for the nominal 4-D hyperchaotic system, then the uncertainty is introduced to the slave system as in (6.3) for different percentages of uncertainty.

$$a1 = a1(1 + \delta a1)$$
$$a2 = a2(1 + \delta a2)$$
(6.3)

Where $\delta$ is the percentage of uncertainty added to the parameters.

To better understand the effect of parameter uncertainty on the synchronization. Fig 19 depicts the synchronization of $x_2$ state variable using Adaptive control and LQR for different

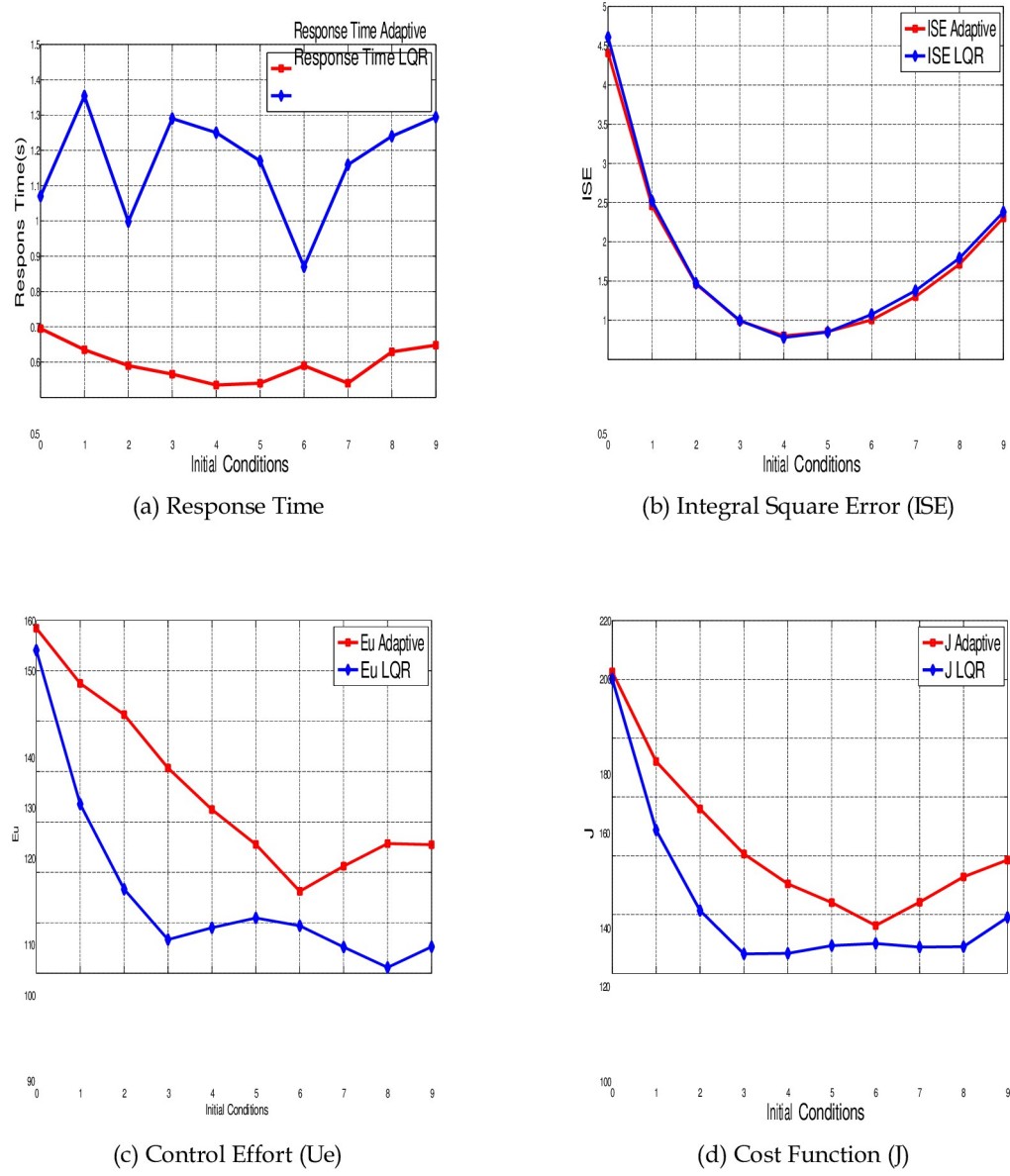

(a) Response Time

(b) Integral Square Error (ISE)

(c) Control Effort (Ue)

(d) Cost Function (J)

**Fig 18. Comparing the performance indicators of LQR and adaptive control in synchronization of the 4-D hyperchaotic system for different values of initial conditions.** (a) Response Time, (b) Integral Square Error (ISE), (c) Control Effort (Ue), and (d) Cost Function (J).

percentages of uncertainty i.e. $\delta$ (varying from 1% till 100%). We focus on the second state variables in Eqs (2.1) and (4.1) since the uncertainty affects it directly.

In Fig 19A and 19B when $\delta$ is less or equal to 40%, the slave signal synchronized the master signal with excellent performance for both control methods since they drive the synchronization error to zero. When $\delta$ increases to 60%, Fig 19C and 19D reveal that Adaptive control is better since the LQR shows deviation from $x_2$ state variable. When $\delta$ increases to reach 100% both controllers show wide deviations from $x_2$ state variable as shown in Fig 19E and 19F.

The performance indicator used for the comparison between the LQR and Adaptive control, in case of parameter uncertainty, is only the integral square error of error-index (ISE) since it is the most significant criterion when the system is subjected to uncertainty.

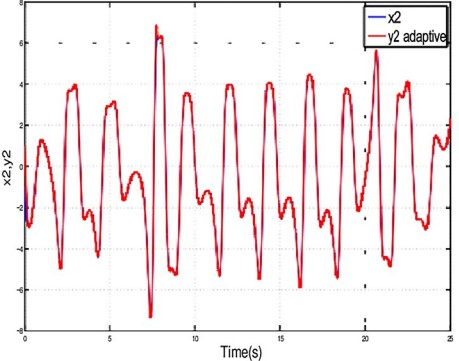
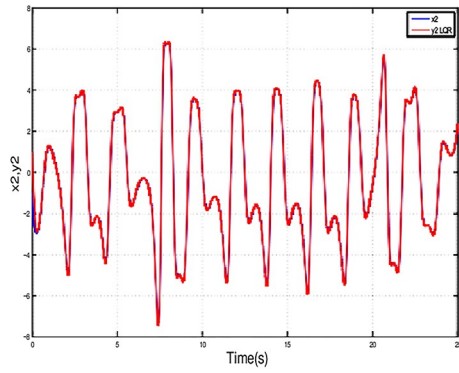

(a) Adaptive control when $\delta = 40\%$        (b) LQR when $\delta = 40\%$

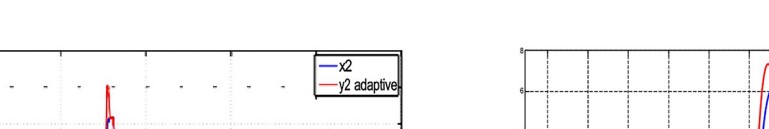
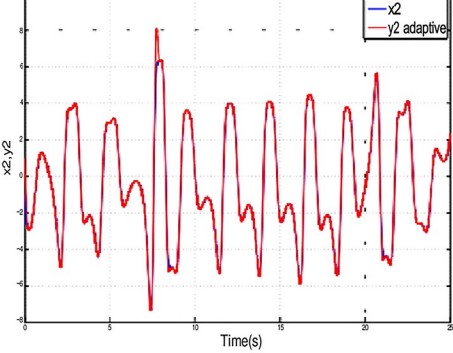
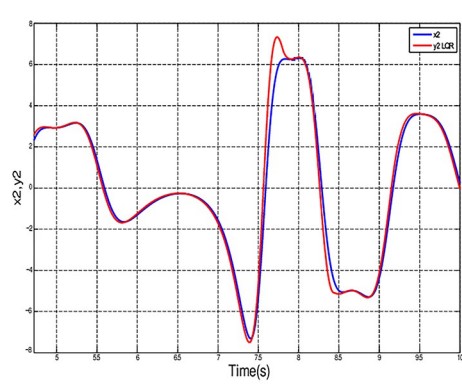

(c) Adaptive control when $\delta = 60\%$        (d) LQR when $\delta = 60\%$

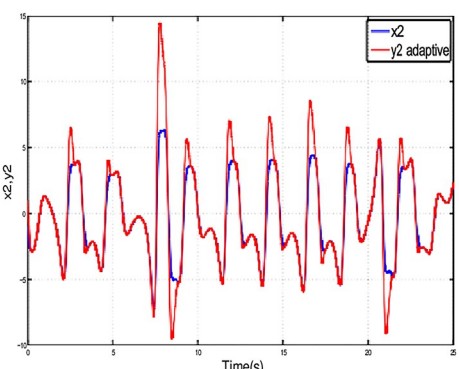
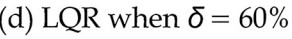
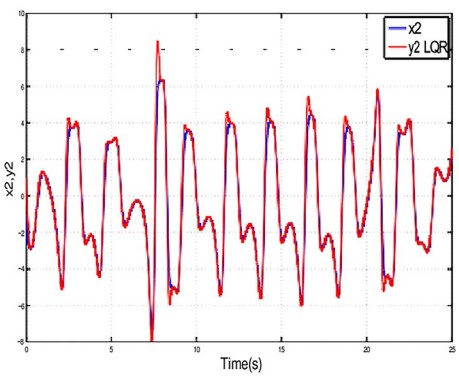

(e) Adaptive control when $\delta = 100\%$        (f) LQR when $\delta = 100\%$

**Fig 19. Time history for $x_2$ synchronization using LQR and adaptive control for different percentages of uncertainty.** (a) Adaptive control when $\delta = 40\%$, (b) LQR when $\delta = 40\%$, (c) Adaptive control when $\delta = 60\%$, (d) LQR when $\delta = 60\%$, (e) Adaptive control when $\delta = 100\%$, and (f) LQR when $\delta = 100\%$.

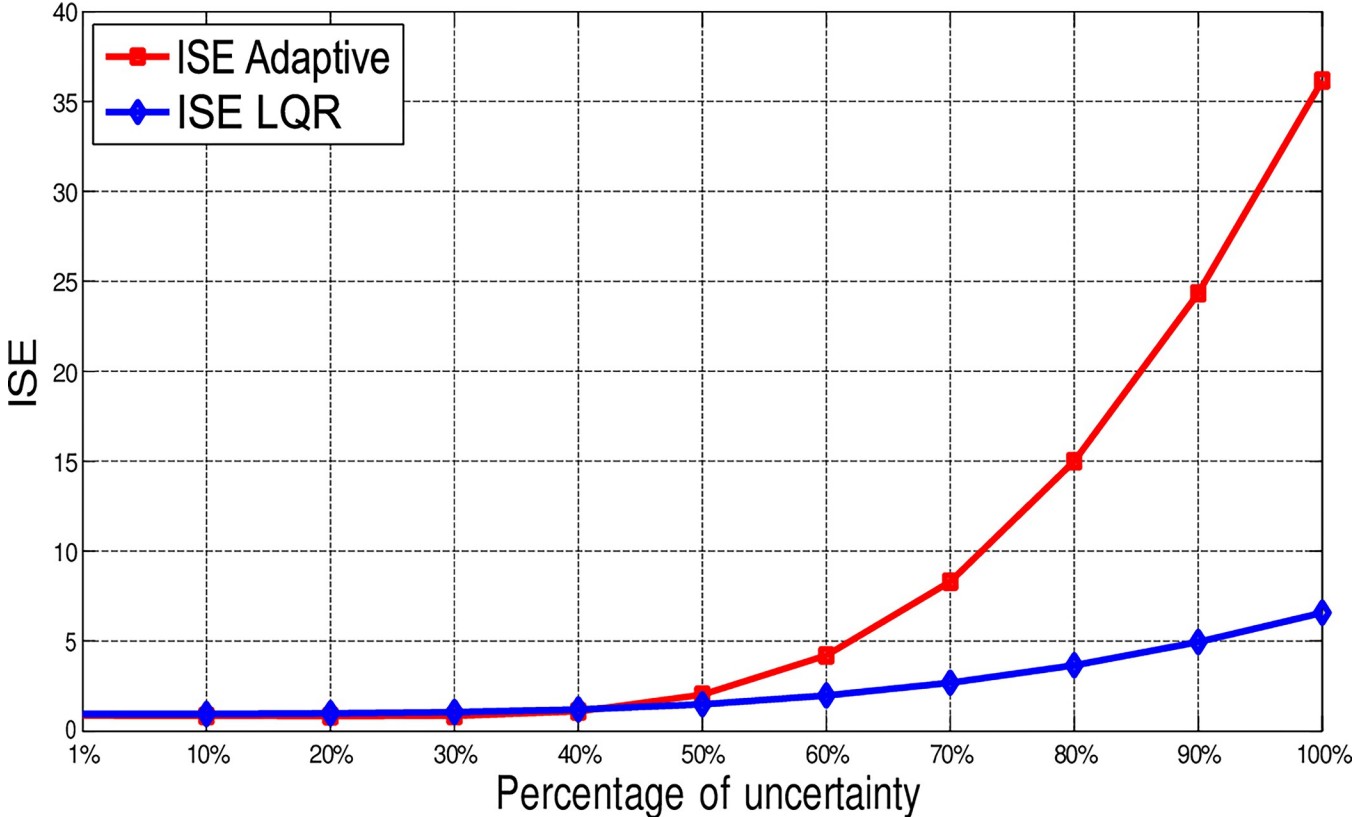

**Fig 20. Comparing the error-index ISE of LQR and adaptive control in synchronization of the 4D-hyperchaotic system for different percentages of uncertainty.**

The ISE is calculated for different percentages of uncertainty. Fig 20 illustrates the graphical representation of ISE for both controllers. When $\delta$ is less than 40%, the error values are almost the same for both controllers. Then by increasing the uncertainty, this error increases rapidly for Adaptive control and slowly for the LQR.

## 7. Synchronization of 3-D chaos-based communication system

In this section, we will apply both synchronization techniques to the 3-D chaotic system in order to draw conclusions about the results of the comparison while also ensuring that the outcomes are consistent across various kinds of chaotic systems. The Proposed 3-D system is a seven-term chaotic oscillator with self-excited attractors and three quadratics nonlinearities. The nonlinear equations of the 3-D chaotic system are given in (7.1).

$$
\begin{aligned}
\dot{x}_1 &= b_1(x_2 - x_1) + b_3 x_2 x_3 \\
\dot{x}_2 &= b_2 x_1 - x_1 x_3 \\
\dot{x}_3 &= x_1 x_2 - x_3
\end{aligned}
\tag{7.1}
$$

Where $x_1$, $x_2$, $x_3$ are state variables and $b_1$, $b_2$, $b_3$ are positive parameters. The system exhibits a chaotic behavior when the parameters $b_1$, $b_2$, and $b_3$ have the values: $b_1 = 10$, $b_2 = 15$, and $b_3 = 12$. The Lyapunov exponents (LE) of the 3-D chaotic system are obtained as $(L_1, L_2, L_3) = (2.71916, 0, -13.72776)$. The first positive LE and chaotic attractors reveal the chaotic behavior of the proposed system.

Fig 21A–21C show the chaotic attractors of the 3-D chaotic oscillator system. The parameter values for the 3-D chaotic system are $b_1=10$, $b_2=15$, and $b_3=12$, and the initial conditions are $x(0) = [5.2, 8.3, -2.7]^T$.

The synchronized 3-D chaotic system is built-in Matlab/Simulink using both control methods, Adaptive control and LQR. Then the comparison is applied for different initial conditions and uncertainty percentages.

## 7.1 Variation in initial conditions

Table 2 shows the numerical representation of performance indicators for synchronization of two identical 3-D chaotic systems defined in (7.1) using both control methods, Adaptive control and LQR.

The first column in Table 2 refers to the case number, the columns (2 to 4) indicate the values of initial conditions, the rest of columns show the values of performance indexes for both control techniques and the last two rows contain the average values for each column and the percentage of the difference between the LQR and Adaptive control for all the indicators. For the adaptive controller of the 3-D chaotics system, we take the gain constants as $k_i = 5$ for $i = 1$, 2, 3. For the LQR, we choose $Q = 100.I_{4\times4} R = I_{4\times4}$. Thus, the control law could be achieved by solving the Ricatti equation in MATLAB using the 'lqr' command to determine P and K matrices.

$$P = K = \begin{bmatrix} 9.7626 & 9.1315 & 0.0000 \\ 9.1315 & 14.1154 & 0.0000 \\ 0.0000 & 0.0000 & 9.0499 \end{bmatrix}$$

Fig 22A–22D describe the values of performance indicator $t_r$, ISE, Ue and J, respectively shown in Table 2 using both control techniques for different values of initial conditions. The blue color refers to LQR and the red one refers to Adaptive control.

Fig 22A shows that the response time of the Adaptive control is faster than the LQR for all values of initial conditions. The minimum value of $t_r$ for the LQR is 1.044 s (at case number 1, where the initial conditions are $y(0) = [-9 - 8 - 7]^T$). For the Adaptive control, the minimum time is 0.55 s (at case number 5, where the initial conditions are $y(0) = [-7 - 6 - 5]^T$). While the average values of response time for all ten different cases are 1.1513 s and 0.679 s for LQR and Adaptive control, respectively. This shows that the speed response of Adaptive control is 41.02% higher than the LQR.

The ISE for both control methods is shown in Fig 22B. The values of the ISE are almost the same for both controllers through all different initial conditions. The average values of ISE are 31.00 and 31.69 for LQR and Adaptive control respectively.

Fig 22C shows the control effort for each controller through different initial conditions. As shown in this figure, the LQR requires energy less than the Adaptive control for all proposed initial conditions. The minimum value for the energy applied from LQR is 144.7780 (in case number 1). For the Adaptive control, the minimum energy is 150.9226 (in case number 1). The average values of energy are 188.6040 and 191.2430 for LQR and Adaptive control, respectively. This shows that energy consumption by LQR is 1.38% more than Adaptive control.

Fig 22D indicates the values of the cost function for both control methods. As shown in this figure, the LQR requires less cost than Adaptive control. The minimum value for the cost indicator related to the LQR is 453.1980 (in case number 1). For the Adaptive control, the minimum cost is 465.1768 (in case number 1). The average values of the cost function are 498.65 and 508.1470 for LQR and Adaptive control, respectively. The returns of the cost function for LQR is also better than Adaptive control by 1.966%.

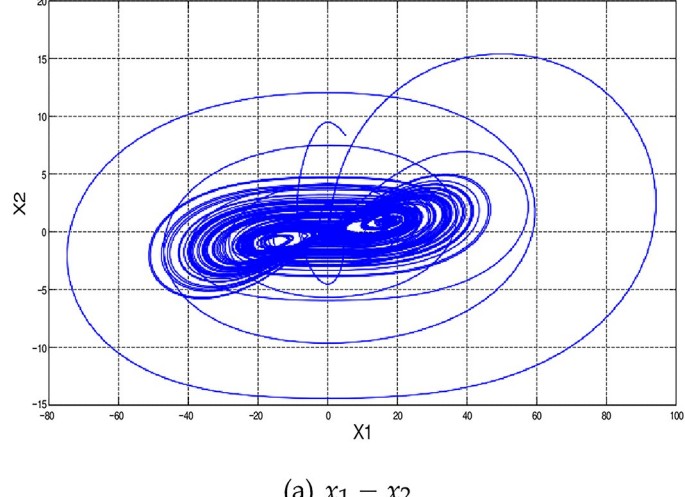

(a) $x_1 - x_2$

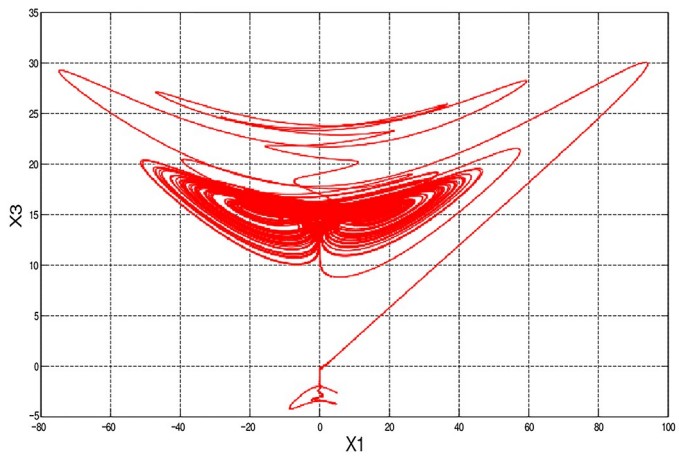

(b) $x_1 - x_3$

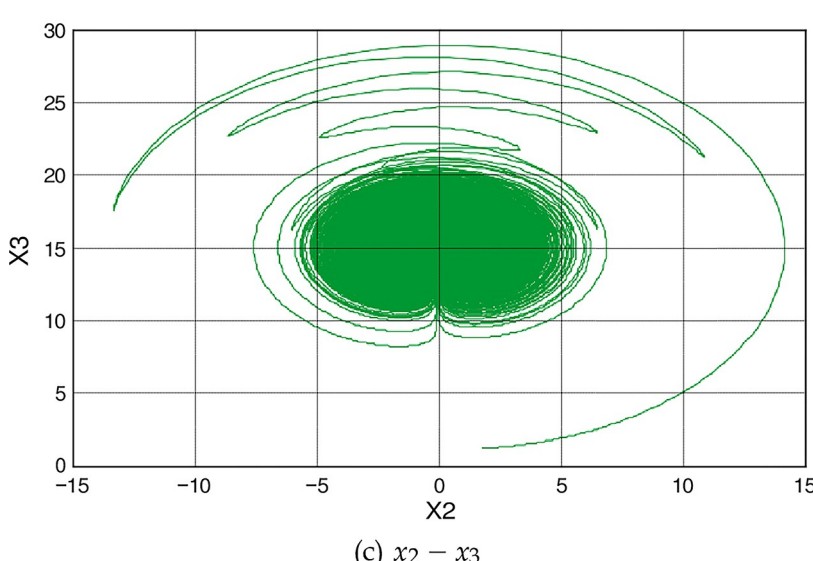

(c) $x_2 - x_3$

**Fig 21. Attractors of the 3-D chaotic oscillator system.** (a) $x_1 - x_2$, (b) $x_1 - x_3$, and (c) $x_2 - x_3$.

**Table 2. The performance indicators for the synchronized 3-D chaotic system using adaptive control and LQR for different values of initial conditions.**

| Case Number | Initial Conditions | | | Response Time (S) | | ISE | | Ue | | J | |
|---|---|---|---|---|---|---|---|---|---|---|---|
| | Y1(0) | Y2(0) | Y3(0) | LQR | Adaptive Control | LQR | Adaptive Control | LQR | Adaptive Control | LQR | Adaptive Control |
| 1 | -9 | -8 | -7 | 1.044 | 0.55 | 30.8420 | 31.4254 | 144.7780 | 150.9226 | 453.1980 | 465.1768 |
| 2 | -8.5 | -7.5 | -6.5 | 1.109 | 0.65 | 31.6359 | 32.3179 | 172.8861 | 169.3827 | 489.2455 | 492.5620 |
| 3 | -8 | -7 | -6 | 1.17 | 0.67 | 32.6394 | 32.9885 | 186.9941 | 183.6314 | 513.3886 | 513.5160 |
| 4 | -7.5 | -6.5 | -5.5 | 1.17 | 0.68 | 33.0295 | 33.4117 | 195.7682 | 197.5823 | 526.0630 | 531.6996 |
| 5 | -7 | -6 | -5 | 1.17 | 0.69 | 32.9890 | 33.5097 | 199.8911 | 203.2121 | 529.7807 | 538.3091 |
| 6 | -6.5 | -5.5 | -4.5 | 1.17 | 0.71 | 32.2976 | 33.0450 | 199.7872 | 203.7521 | 522.7635 | 534.2021 |
| 7 | -6 | -5 | -4 | 1.17 | 0.71 | 31.1890 | 32.2304 | 199.0696 | 203.9787 | 510.9594 | 526.2832 |
| 8 | -5.5 | -4.5 | -3.5 | 1.17 | 0.71 | 30.0865 | 31.1797 | 198.7084 | 204.4309 | 499.5737 | 516.2277 |
| 9 | -5 | -4 | -3 | 1.17 | 0.71 | 28.7333 | 29.5245 | 197.5061 | 202.6325 | 484.8391 | 497.8779 |
| 10 | -4.5 | -3.5 | -2.5 | 1.17 | 0.71 | 26.6127 | 27.2774 | 190.6376 | 192.9158 | 456.7645 | 465.6894 |
| Average | | | | 1.1513 | 0.6790 | 31.0059 | 31.6920 | 188.6040 | 191.2430 | 498.6570 | 508.1470 |
| Differance in percent | | | | 41.02% | | 2.165% | | 1.38% | | 1.966% | |

## 7.2 Parameter uncertainty

To illustrate the effect of parameter uncertainty on the synchronization. Fig 23 shows the synchronization of $x_1$ state variable using Adaptive control and LQR for different percentages of uncertainty. We focus on $x_1$ since the uncertainty affects it directly.

In Fig 23A and 23B when $\delta$ is less or equal to 20%, the slave signal synchronized the master signal perfectly since the synchronization error converges to zero. By increasing the percentage of uncertainty the LQR introduces a small deviation. Increasing $\delta$ to 40% causes a small deviation in the Adaptive control as shown in Fig 23C. With this uncertainty, the adaptive method is still better than the LQR since the synchronization error is still less (see Fig 23D). When $\delta$ increases to reach 100% both controllers show wide deviations from $x_1$ state variable as shown in Fig 23E and 23F.

Fig 24 shows the return of the error-index (ISE) for both LQR and Adaptive control in synchronization of the proposed 3-D chaotic system for different percentages of uncertainty. When $\delta$ is less than 10%, the error-index (ISE) for both control methods are almost the same. This value rises when $\delta$ increases until 30%. However, until this point, the return of Adaptive control is better since the LQR introduces some deviations. By increasing $\delta$ gradually to 100% even the Adaptive control introduces small error, the values of ISE related to both controllers increase significantly.

## 8. Results discussion

Table 3 compares the performance indicators between Adaptive control and LQR for the 3-D chaotic and 4-D hyperchaotic systems. As shown in the previous sections and summarized in Table 3, the results obtained for synchronizing the chaos-based secured communication system is independent from the chaotic oscillator used to build the system. Hence, the results for the 4-D hyperchaotic system are identical to the 3-D chaotic system. The results also show that the two control synchronization approaches (Adaptive control and LQR) converge the synchronization error asymptotically to zero regardless of the values of initial conditions.

When the response time is considered, the Adaptive control shows a faster response than the LQR by an average of 48.97% for the hyperchaotic system and 41.02% for the 3-D chaotic system. Regarding the integral square error (ISE), the values are very similar for both

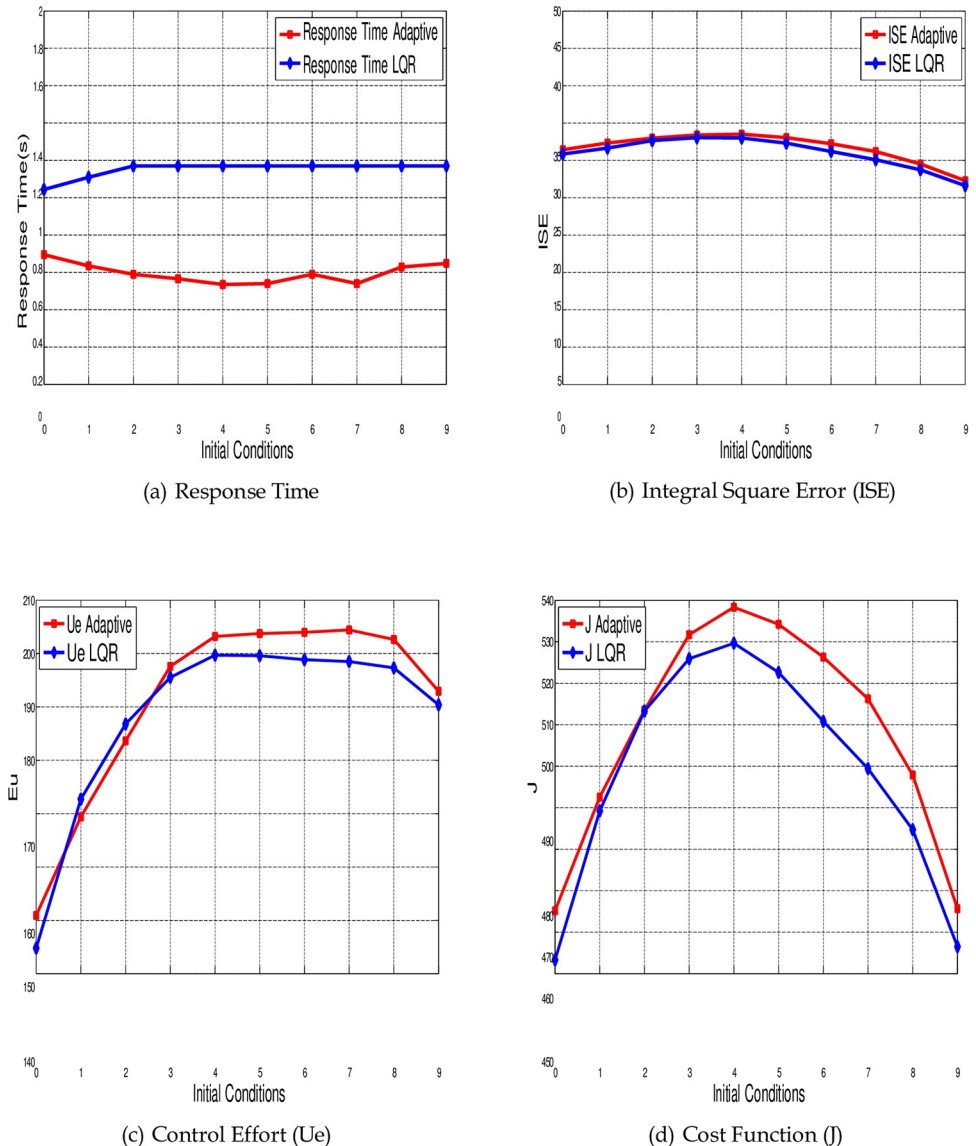

**Fig 22. Comparing the performance indicators of LQR and adaptive control in synchronization of the 3-D chaotic system for different values of initial conditions.** (a) Response Time, (b) Integral Square Error (ISE), (c) Control Effort (Ue), and (d) Cost Function (J).

controllers. The average errors are 3.09% and 2.165% for the hyperchaotic and chaotic systems respectively. When the control effort (Ue) is considered, the LQR shows less energy consumption than the adaptive method by an average of 16.03% for the hyperchaotic system and by 1.38% for the chaotic system. The cost function (J) for the LQR is less on average by 13.72% and 1.966% for the hyperchaotic and chaotic systems, respectively.

Considering the uncertainty, if the percentage of uncertainty is in the specific range (40% for the hyperchaotic system and 20% for the 3-D chaotic system), both controllers could achieve synchronization and make slave states follow master states perfectly when time passes since the synchronization error tend to zero, However, in this specific range, the LQR shows slightly better results (less ISE) than the Adaptive control. On the other hand, the adaptive method can still make synchronization for a wider range.

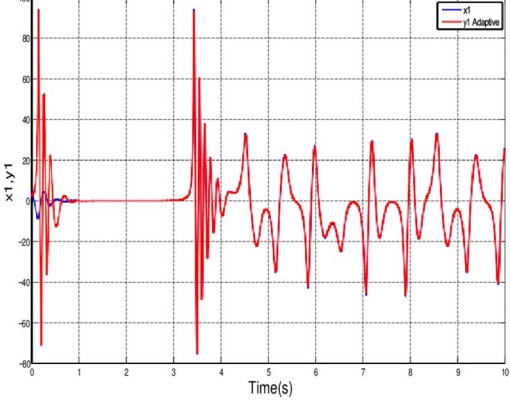

(a) Adaptive control when $\delta = 20\%$

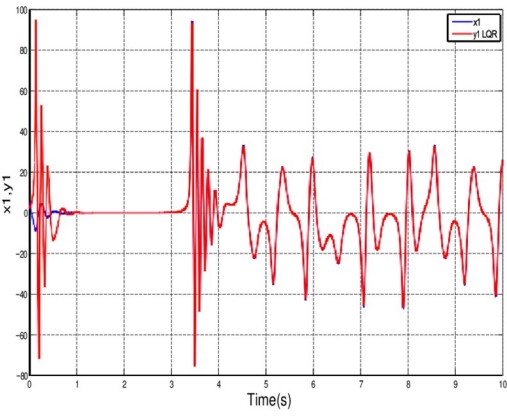

(b) LQR when $\delta = 20\%$

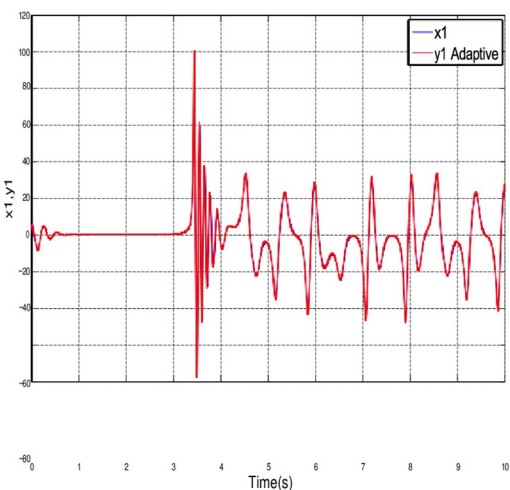

(c) Adaptive control when $\delta = 40\%$

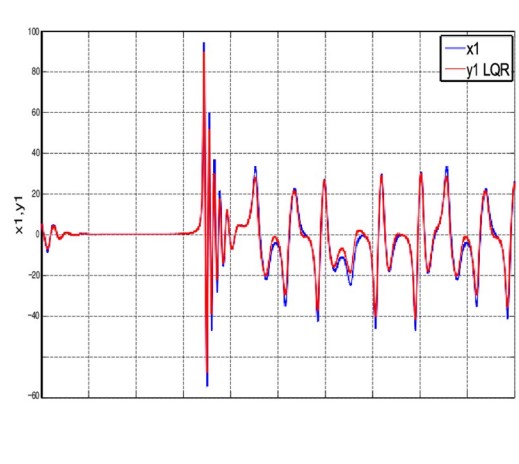

(d) LQR when $\delta = 40\%$

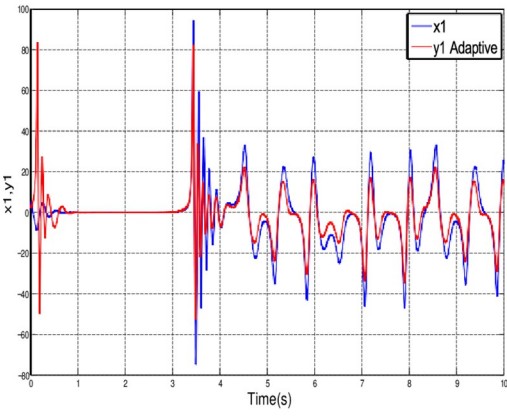

(e) Adaptive control when $\delta = 100\%$

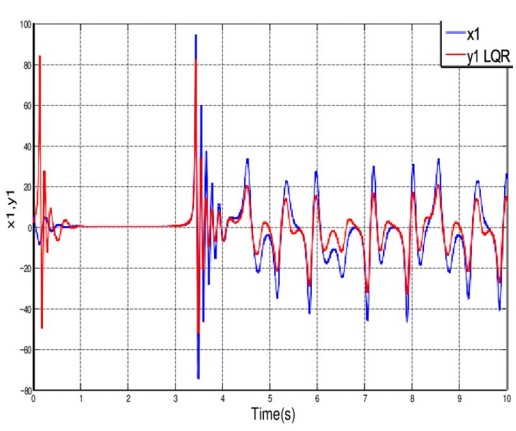

(f) LQR when $\delta = 100\%$

**Fig 23. Time history for $x_1$ synchronization using LQR and adaptive control for different percentages of uncertainty.** (a) Adaptive control when $\delta$ = 20%, (b) LQR when $\delta$ = 20%, (c) Adaptive control when $\delta$ = 40%, (d) LQR when $\delta$ = 40%, (e) Adaptive control when $\delta$ = 100%, and (f) LQR when $\delta$ = 100%.

Table 4 shows the FPGA resources utilization and clock frequency for each system block including the master system, slave system, adaptive controller, and LQR controller. As shown in the table, the slave system requires more FPGA resources of Flip-Flops (FF), Look-up Tables (LUT), and DSPs, than the master system as it has more complex equations. The frequency of both systems is the same 102.155 MHz.

Table 4 also shows a comparison between the Adaptive and the LQR controllers. The FPGA resources required to implement the LQR controller is less than the Adaptive one. It requires 47 less FFs, 235 less LUTs, and 20 less DSPs. In addition to that, the frequency of LQR controller is 34 MHz higher than the adaptive one as it requires less number of basic blocks (see sections 5.3 and 5.4).

Table 5 shows the FPGA resources utilization and clock frequency of a communication system when Adaptive controller is used and then when LQR controller is used. As the LQR controller utilizes less FPGA resources than the Adaptive one (as shown in Table 4), the complete communication system synchronized using LQR controller requires less FPGA resources than the Adaptive one. Moreover, its clock frequency is 103 MHz which is 21 MHz higher than the Adaptive controller.

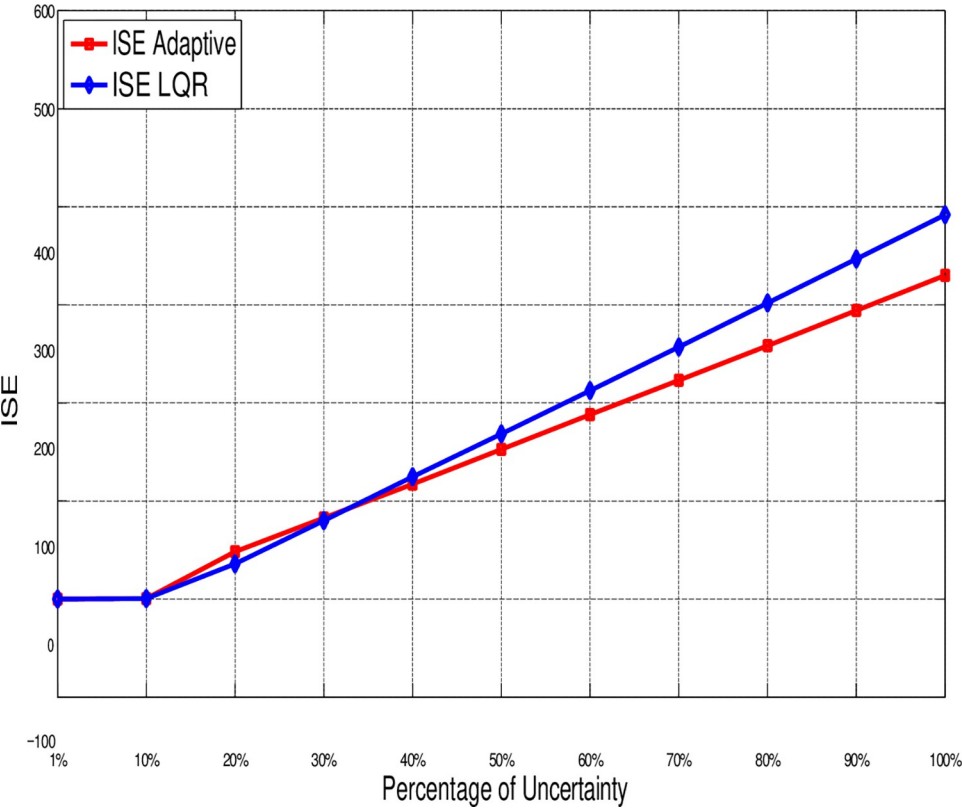

**Fig 24. Comparing the error-index ISE of LQR and adaptive control in synchronization of the 3-D chaotic system for different percentages of uncertainty.**

**Table 3. Comparing the performance indicators between adaptive control and LQR for the 3-D chaotic and 4-D hyperchaotic systems.**

| Performance Indicators | 3-D Chaotic System | | 4-D Hyperchaotic System | |
|---|---|---|---|---|
| | Adaptive | LQR | Adaptive | LQR |
| Response Time (tr) | ✓ | | ✓ | |
| Integral Square Error (ISE) | ≈ | ≈ | ≈ | ≈ |
| Control Effort (Ue) | | ✓ | | ✓ |
| Cost Function (J) | | ✓ | | ✓ |
| Uncertainty in Specific Range | | ✓ | | ✓ |
| Wider Uncertainty | ✓ | | ✓ | |
| Easy to Design | | ✓ | | ✓ |

**Table 4. FPGA resources utilization and clock frequency for each system block.**

| | Master Block | | Slave Block | | Adaptive Block | | LQR Block | |
|---|---|---|---|---|---|---|---|---|
| | Units | Utilization % | Units | Utilization % | Units | Utilization % | Units | Utilization % |
| FF | 603 | 0.57 | 731 | 0.69 | 730 | 0.69 | 683 | 0.64 |
| LUT | 608 | 1.14 | 802 | 1.51 | 1229 | 2.31 | 994 | 1.87 |
| DSP48 | 18 | 8.18 | 18 | 8.18 | 56 | 25.45 | 36 | 16.36 |
| CLK Frequency (MHz) | 102.155 | | 102.155 | | 97.343 | | 131.062 | |

**Table 5. FPGA resources utilization and clock frequency of the communication system with different controllers.**

| | Communication System with Adaptive Controller | | Communication System with LQR Controller | |
|---|---|---|---|---|
| | Units | Utilization % | Units | Utilization % |
| FF | 2129 | 2.00 | 2062 | 1.94 |
| LUT | 2511 | 4.72 | 2333 | 4.39 |
| DSP48 | 92 | 41.82 | 72 | 32.73 |
| CLK Frequency (MHz) | 82.008 | | 103.831 | |

In conclusion, the response time of the Adaptive control is faster than the LQR and can achieve synchronization for a wide range of parameter uncertainty. However, the LQR control method is much better in the sense of state energy and cost function criterion. Moreover, implementing it on FPGA is much simpler and requires less FPGA resources than the adaptive controller which demands complex computations. Finally, the LQR is robust enough to overcome the uncertainty within a specific range depending on the system.

## 9. Conclusion

This paper studied the synchronization for two different chaotic systems: 4-D hyper-chaotic and 3-D chaotic systems, using two different control approaches: Adaptive non-linear controller and linear optimal quadratic regulator (LQR). The two controller methods were compared based on different performance indicators. Then the synchronization approaches were realized on the FPGA platform using VHDL code. The simulations and calculation show that the LQR method is more efficient than the Adaptive control method from the perspective of energy consumption, cost function, and parameter uncertainty in a specific range. The hardware implementation on FPGA shows that the LQR approach requires fewer resources utilization than the adaptive approach. Moreover, it is much simpler to design since it does not need

complex calculations. However, Adaptive control is much faster in synchronization and has a wide range of parameter uncertainty. Also the adaptive control operates in the lower frequency when applied on the FPGA platform. In the future work, we will implementation hardware of secure communication system.

## Supporting information

**S1 File.**
(RAR)

**S2 File.**
(RAR)

## Author Contributions

**Conceptualization:** Talal Bonny.

**Formal analysis:** Talal Bonny.

**Funding acquisition:** Talal Bonny.

**Methodology:** Talal Bonny.

**Project administration:** Talal Bonny.

**Resources:** Talal Bonny.

**Validation:** Wafaa Al Nassan, Aceng Sambas.

**Visualization:** Wafaa Al Nassan, Aceng Sambas.

**Writing – original draft:** Wafaa Al Nassan, Aceng Sambas.

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
