## [Decision Letter · Decision Letter 0]

26 Jul 2023

PONE-D-23-16430Comparative Analysis and FPGA Realization of Different Control Synchronization Approaches for Chaos-based Secured Communication SystemsPLOS ONE

Dear Dr. Bonny,

Thank you for submitting your manuscript to PLOS ONE. After careful consideration, we feel that it has merit but does not fully meet PLOS ONE’s publication criteria as it currently stands. Therefore, we invite you to submit a revised version of the manuscript that addresses the points raised during the review process.

We look forward to receiving your revised manuscript.

Kind regards,

Jun Ma, Dr.

Academic Editor

PLOS ONE

Journal Requirements:

Reviewers' comments:

Reviewer's Responses to Questions

**Comments to the Author**

1. Is the manuscript technically sound, and do the data support the conclusions?

Reviewer #1: Yes

2. Has the statistical analysis been performed appropriately and rigorously? 

Reviewer #1: Yes

3. Have the authors made all data underlying the findings in their manuscript fully available?

Reviewer #1: Yes

4. Is the manuscript presented in an intelligible fashion and written in standard English?

Reviewer #1: Yes

5. Review Comments to the Author

Reviewer #1: This paper clearly demonstrates the incorrect signal recovery caused by variations of initial conditions and state parameters, proposes two different control approaches with stability analysis and performance comparison stating the pros and cons. I would suggest publication of this paper after solving my following questions:

(1) Please fix typos and format incompatibility issues, such as question mark at citation number 8 and equation 4.10 (derivative of e instead of e).

(2) I also published one analog implementation (https://dergipark.org.tr/en/download/article-file/2772051) of a nonlinear system demonstrating chaos via FPAA, is there any reasons why you choose digital system implementation instead of analog system implementation?

(3) Your variable u is used to represent both control inputs and system states, which is very confusing, I would recommend changing them to two different representations.

(4) Could you show some simulation results comparing the performance between adaptive controller and LQR controller? As I could not find it anywhere in your papers.

(5) Your controller is initially designed to work with a 4-D system, and then you used a 3-D system to demonstrate the robustness of your proposed controllers, while, how can you directly apply the same controller to two different systems with different dimensions? Could you explain more details on this?

6. PLOS authors have the option to publish the peer review history of their article (what does this mean?). If published, this will include your full peer review and any attached files.

Reviewer #1: **Yes: **XiaoFu Li

---

## [Author Response · Author response to Decision Letter 0]

17 Aug 2023

Dear Editor

We addressed all reviewer's comments and submitted the response file

---

## [Decision Letter · Decision Letter 1]

6 Sep 2023

Comparative Analysis and FPGA Realization of Different Control Synchronization Approaches for Chaos-based Secured Communication Systems

PONE-D-23-16430R1

Dear Dr. Bonny,

We’re pleased to inform you that your manuscript has been judged scientifically suitable for publication and will be formally accepted for publication once it meets all outstanding technical requirements.

Kind regards,

Jun Ma, Dr.

Academic Editor

PLOS ONE

Reviewer #1: The author has solved each one of my concerns and questions properly, and I am satisfied with his/her answers and I would recommend publication of this paper.

Reviewer #1: **Yes: **XiaoFu Li

---

## [Editor Report · Acceptance letter]

14 Sep 2023

PONE-D-23-16430R1 

Comparative Analysis and FPGA Realization of Different Control Synchronization Approaches for Chaos-based Secured Communication Systems 

Dear Dr. Bonny:

I'm pleased to inform you that your manuscript has been deemed suitable for publication in PLOS ONE. Congratulations! Your manuscript is now with our production department. 

Kind regards, 

on behalf of

Dr. and Pro. Jun Ma 

Academic Editor

PLOS ONE